# Toll-like receptor 4-mediated inflammation triggered by extracellular IFI16 is enhanced by lipopolysaccharide binding

Andrea Iannucci[1,2], Valeria Caneparo[1,2], Stefano Raviola[1,2], Isacco Debernardi[2], Donato Colangelo[3], Riccardo Miggiano[4], Gloria Griffante[2,5], Santo Landolfo[5], Marisa Gariglio[1,2]*, Marco De Andrea[1,5]*

1 CAAD—Center for Translational Research on Autoimmune and Allergic Disease, University of Eastern Piedmont, Novara, Italy, 2 Department of Translational Medicine, University of Eastern Piedmont, Novara, Italy, 3 Department of Health Sciences, University of Eastern Piedmont, Novara, Italy, 4 Department of Pharmaceutical Sciences, University of Eastern Piedmont, Novara, Italy, 5 Department of Public Health and Pediatric Sciences, University of Turin, Medical School, Turin, Italy

* marisa.gariglio@uniupo.it (MG); marco.deandrea@unito.it (MDA)

**Data Availability Statement:** All relevant data are within the manuscript and its Supporting Information files.

## Abstract

Damage-associated molecular patterns (DAMPs) are endogenous molecules activating the immune system upon release from injured cells. Here we show that the IFI16 protein, once freely released in the extracellular milieu of chronically inflamed tissues, can function as a DAMP either alone or upon binding to lipopolysaccharide (LPS). Specifically, using pull-down and saturation binding experiments, we show that IFI16 binds with high affinity to the lipid A moiety of LPS. Remarkably, IFI16 DAMP activity is potentiated upon binding to sub-toxic concentrations of strong TLR4-activating LPS variants, as judged by TLR4-MD2/TIRAP/MyD88-dependent IL-6, IL-8 and TNF-α transcriptional activation and release in stimulated monocytes and renal cells. Consistently, using co-immunoprecipitation (co-IP) and surface plasmon resonance (SPR) approaches, we show that IFI16 is a specific TLR4-ligand and that IFI16/LPS complexes display a faster stimulation turnover on TLR4 than LPS alone. Altogether, our findings point to a novel pathomechanism of inflammation involving the formation of multiple complexes between extracellular IFI16 and subtoxic doses of LPS variants, which then signal through TLR4.

## Author summary

IFI16 is a nuclear protein involved in a variety of physiological processes, including cell cycle regulation, tumor suppression, and virus sensing. Emerging evidence indicates that IFI16 is released in the extracellular milieu under injury or stress conditions. Here we show that extracellular IFI16 acts as a damage-associated molecular pattern (DAMP), triggering inflammation through Toll-like receptor 4 (TLR4) activation. Furthermore, we demonstrate that IFI16 activity is potentiated upon binding to subtoxic concentrations of strong TLR4-activating lipopolysaccharide (LPS) variants, which are known to be present

**Funding:** The investigators were supported by grant 2015W729WH from the Italian Ministry for University and Research (PRIN2015 to MDA), grant CST0168881 from the Compagnia di San Paolo (CSP2016 to MDA), and grant from the European Commission under the Horizon2020 program (H2020-MSCA-ITN-2015 to SL). This study was partially funded by the AGING Project – Department of Excellence– DIMET, University of Eastern Piedmont. The funders had no role in study design, data collection and analysis, decision to publish, or preparation of the manuscript.

**Competing interests:** The authors have declared that no competing interests exist.

in various pathological settings other than gram-negative infections. Our study provides new insights into the role of extracellular IFI16 during low-grade endotoxemia.

## Introduction

In the absence of stress stimuli, expression of the nuclear IFI16 protein is restricted to hematopoietic cells, vascular endothelial cells and keratinocytes [1]. Since its discovery in the early 90s, IFI16 has been involved in a growing number of physiological processes, such as cell cycle regulation, tumor suppression, apoptosis, DNA damage signaling, virus sensing and virus restriction [2–5]. More recently, we and others have found that IFI16 can also be aberrantly expressed in chronically inflamed tissues such as the intestinal epithelium of patients with inflammatory bowel disease (IBD) [6,7] and the epidermis and inflammatory dermal infiltrates of systemic lupus erythematosus (SLE) patients [8,9]. In addition, abnormal IFI16 expression has been also detected in the skin of individuals affected by systemic sclerosis (SSc) [10] or psoriasis (Pso) [11–13], as well as in salivary epithelial cells and infiltrating lymphocytes of subjects with Sjögren's syndrome (SS) [14,15]. Noteworthy, serum circulating IFI16 protein and its specific autoantibodies have also been reported in various autoimmune diseases, including SSc, rheumatoid arthritis (RA), SLE, SS, psoriatic arthritis (PsA) and IBD [6,8,16–20]. Our group and others have also reported IFI16 de-localization to the cytoplasm upon viral infection or UVB exposure [8,21–23]. Importantly, under these conditions, IFI16 is eventually released in the extracellular matrix where it acts as a damage-associated molecular pattern (DAMP), inducing a proinflammatory phenotype [8,24]. However, the molecular mechanisms underlying the extracellular DAMP activity of IFI16 have yet to be determined.

Lipopolysaccharide (LPS) is the main cause of gram-negative bacterial sepsis and among the best-characterized pathogen-associated molecular patterns (PAMPs). It is made up of a lipid component (lipid A), playing an essential role in promoting inflammation, and two sugar moieties subdivided in a core polysaccharide and an O-polysaccharide of variable length [25]. Serum LPS is recognized by the LPS binding protein (LBP), which then forms transient ternary complexes with soluble or membrane-anchored CD14 (sCD14 or mCD14, respectively). Subsequently, CD14 dissociates from LBP to extract monomeric LPS [26, 27] and through mCD14, LPS is finally presented to the TLR4/MD2 complex, thereby leading to the activation of multiple signaling components, including NF-κB and IRF3, which in turn transcriptionally activate proinflammatory cytokines [28].

Since we previously demonstrated that recombinant IFI16 can synergize with subtoxic concentrations of LPS to induce proinflammatory cytokine production in endothelial cells [24], here we have explored the possibility that aberrant expression of IFI16 and its ensuing release into the extracellular matrix may favor its interaction with exogenous molecules, such as LPS, thereby triggering an inflammatory state also in other target cells.

In this study, we show for the first time that IFI16 binds with high affinity to the lipid A moiety of LPS through its HINB domain. Moreover, we provide further evidence that IFI16 functions as a DAMP by triggering proinflammatory cytokine production in renal and monocytic cell lines through the Toll-like receptor 4 (TLR4) signaling pathway, either alone or, more potently, when complexed with strong TLR4-activating LPS variants.

## Results

### IFI16 binds to LPS of different bacterial origin and inflammatory activity

To investigate the occurrence of a direct association between IFI16 and LPS, we performed an *in vitro* pull-down assay using biotin-labeled LPS from *E. coli* O111:B4 (biotin-LPS-EB) and

human recombinant IFI16 protein. As shown in Fig 1A, we could readily detect a highly reproducible ~100-kDa band corresponding to biotin-LPS-bound IFI16. To rule out that this binding was due to bacterial contaminants, we next performed a pull-down assay using a recombinant glutathione-S transferase (GST) protein prepared with the same procedure as that employed to obtain recombinant IFI16. As shown in Fig 1B, we failed to isolate any GST-containing band following incubation of GST with biotin-LPS-EB and streptavidin beads, demonstrating the specificity of the IFI16/LPS interaction. Furthermore, saturation binding experiments using IFI16-coated microtiter plates challenged with increasing amounts of biotin-LPS-EB revealed biotin-labeled LPS bound to solid-phase IFI16 in a concentration-dependent manner, reaching saturation at 100,000 ng/ml of biotin-LPS-EB (Fig 1C). When recombinant GST or BSA were coated onto the microtiter plates, no binding occurred in the presence of biotin-LPS-EB (Fig 1C). To assess binding specificity, we asked whether polymyxin B (PMB), an LPS-sequestering agent able to bind to negatively charged phosphate groups of lipid A [29], would disrupt IFI16/LPS interaction. As shown in Fig 1C, when PMB was preincubated with biotin-LPS-EB and then added to the IFI16-coated wells, it completely prevented IFI16 from binding to LPS. Next, IFI16/LPS-EB interaction was confirmed by surface plasmon resonance (SPR) analysis flowing increasing amounts of LPS-EB over a CM5 IFI16-coated chip. As shown in Fig 1D, LPS interacted with IFI16 in a concentration-dependent manner, with a kinetic association constant ($K_a$) of $1.13^*10^4$ 1/Ms and a kinetic dissociation constant ($K_d$) of $1.94^*10^{-3}$ 1/s, respectively.

Altogether, these results indicate that IFI16 binds to LPS-EB with high affinity and that such interaction is inhibited by PMB, presumably by masking the negatively charged groups of the LPS lipid A moiety.

We next sought to determine whether IFI16 could bind to LPS in its natural setting, such as the outer membrane of fixed gram-negative bacteria. To this end, a panel of gram-negative bacteria, including a laboratory strain of *E. coli* and a clinical isolate of *Klebsiella pneumoniae*, were assessed as solid phase antigens by whole cell ELISA. Gram-positive clinical isolates of *Staphylococcus aureus*, *Staphylococcus epidermidis* and *Streptococcus pyogenes* were used as negative controls. As shown in Fig 1E, IFI16 strongly associated with the surface of both gram-negative bacteria species. By contrast, no IFI16 binding could be detected when gram-positive bacteria were used as solid phase antigens (Fig 1E). Thus, IFI16-LPS binding can also occur in the natural setting where LPS is anchored to the bacterial outer membrane by its lipid A moiety.

We next asked whether IFI16 would bind with the same affinity to LPS variants derived from different gram-negative strains with highly variable structure and broad-spectrum activity. For this purpose, microtiter plates were coated with the following LPS variants: 1) the two full TLR4 agonists *E. coli* O111:B4 (LPS-EB) and *E. coli* F583 (LPS-F583), with the latter harboring a similar lipid A moiety but a shorter polysaccharide chain length compared to that of the O111:B4 strain; 2) the weak TLR4 agonist *P. gingivalis* (LPS-PG), carrying a mixture of di-, mono- and de-phosphorylated penta- or tetra-acylated lipid A moieties; or 3) the TLR4 antagonist *R. sphaeroides* (LPS-RS), harboring a di-phosphorylated lipid A loaded with 3 long and 2 short acyl chains (Fig 1F). As shown in Fig 1G, IFI16 was able to bind to all the aforementioned solid-phase LPS variants in a concentration-dependent manner, although with slightly different kinetics. Specifically, we obtained similar $K_D$s for the two *E. coli* LPS variants—*i.e.*, 4.2 nM and 4.3 nM for LPS-EB and LPS-F583, respectively—, while we observed slightly higher $K_D$s for LPS-PG and LPS-RS—*i.e.*, 12.0 nM and 19.3 nM, respectively (Table 1). Consistently, treatment of immobilized LPS molecules with PMB prior to the addition of IFI16 completely abolished IFI16 binding. Thus, IFI16 binds to not only the canonical TLR4-activating LPS but also variants characterized by weaker triggering activity.

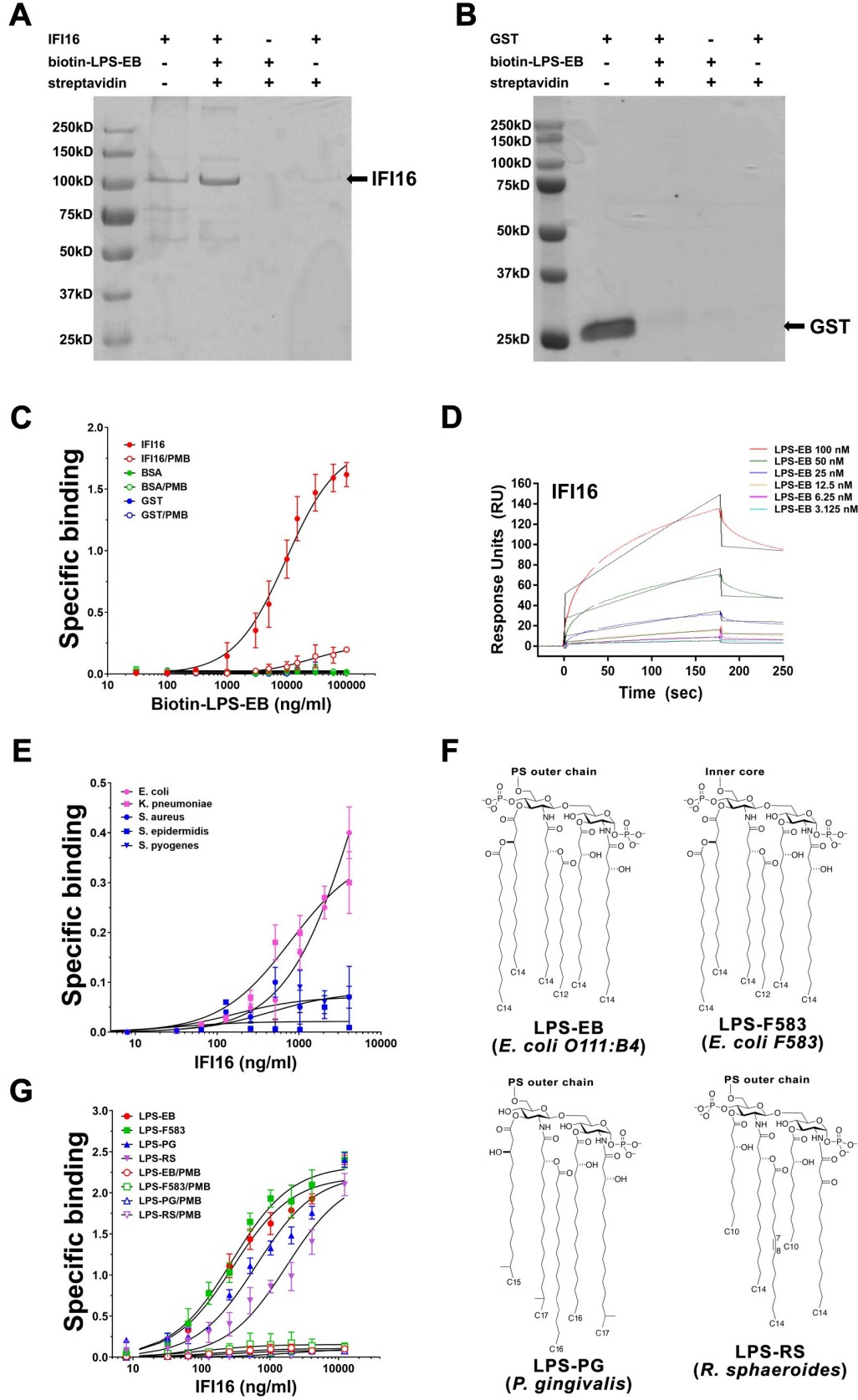

**Fig 1. IFI16 binds to LPS of different bacterial origin and inflammatory activity.** Coomassie brilliant blue staining of pull-down assays performed with 3 μg of recombinant IFI16 (**A**) or GST (**B**) in the presence or absence of biotin-labeled lipopolysaccharide (LPS) from *E. coli* O111:B4 (biotin-LPS-EB). (**C**) Saturation binding experiments performed with 2 μg/ml of IFI16 (red circles) and increasing amount of biotin-LPS-EB. Binding was detected by ELISA using HRP-conjugated streptavidin. Optical density (OD) of samples was measured at 450 nm. An excess of recombinant GST (blue circles) or BSA (green circles) and pre-treatment of biotin-LPS-EB with polymyxin B (PMB, empty circles) were used as negative controls. Data are expressed as mean values ± SD of three independent experiments. (**D**) Surface plasmon resonance (SPR) analysis of LPS-EB binding to immobilized IFI16. After immobilization of IFI16 on the CM5 sensor chip surface, increasing concentration of LPS-EB (3.125–100 nM) diluted in running buffer were injected over immobilized IFI16. Data are representative of three independent experiments. (**E**) *Ex-vivo* interaction analysis between increasing amount of recombinant IFI16 and formalin-fixed gram-negative (*E. coli* and *K. pneumonia*; pink circles and pink squares, respectively) or gram-positive (*S. aureus*, *S. epidermidis*, *S. pyogenes*; blue circles, squares and triangles, respectively) bacteria. Data are expressed as mean values ± SD of three independent experiments. (**F**) Lipid A structures of LPS derived from *E. coli* O111:B4 or F583 LPS (LPS-EB and LPS-F583, respectively; strong TLR4 agonists), *P. gingivalis* (LPS-PG; weak TLR4 agonist) and *R. sphaeroides* (LPS-RS, TLR4 antagonist). For LPS-PG, which harbors a mixture of di-, mono- and de-phosphorylated penta- or tetra-acylated lipid A moieties, a single isoform is represented for simplicity. PS-outer chain = polysaccharide outer chain. (**G**) Saturation binding experiments with increasing amount of recombinant IFI16 (8 to 12,288 ng/ml) and 10 μg/ml of LPS-EB (red line), LPS-F583 (green line), LPS-PG (blue line) or LPS-RS (purple line). Anti-IFI16 antibodies against the N-terminus of the protein and HRP-labelled anti-rabbit IgG were added as primary and secondary antibody, respectively, and binding was detected by ELISA at 450 nm. Data are expressed as mean values ± SD of three independent experiments.

## IFI16 binds to the lipid A moiety of LPS through its HINB domain

To identify which LPS moiety is involved in IFI16 binding, we performed saturation binding experiments using two different variants of lipid A derived from the *E. coli* F583 strain, namely diphosphorylated and monophosphorylated lipid A (DPLA and MPLA, respectively), alongside a detoxified LPS molecule derived from the *E. coli* strain O111:B4 (detoxLPS) (Fig 2A). The first two molecules lack the heteropolysaccharide outer chain and differ in the number of phosphate groups, with MPLA being a weaker agonist than DPLA [30]. On the other hand, the detoxLPS lipid A moiety is partially delipidated by alkaline hydrolysis, resulting in only four primary acyl chains being directly esterified with the sugar moiety, in which the outer chain is however preserved. DetoxLPS endotoxin levels are about 10,000 times lower than that of parental LPS [31]. As shown in Fig 2B, IFI16 readily bound to both forms of lipid A in a concentration-dependent fashion. The $K_D$ values showed higher affinity for the lipid A moieties (either form) in comparison with LPS-F583–0.9 nM and 1.2 nM, respectively, *vs*. 4.3 nM (Table 1). Interestingly, the $K_D$ value for IFI16 binding to detoxLPS (55.6 nM) was the highest among all LPS forms, indicating that the canonical acyl chain is required for IFI16 binding to LPS. When lipid A was pre-treated with PMB, no signal was detected. Thus, LPS binds to IFI16 through its lipid A moiety. To corroborate these data, a competition ELISA was performed by immobilizing LPS from the *E. coli* strain O111:B4 (LPS-EB) onto the microtiter

**Table 1. Full-length IFI16 and IFI16 domains binding affinities to LPS.**

| LPS variant | Equilibrium dissociation constant ($K_D$), nM | | | |
|---|---|---|---|---|
| | IFI16 | PYRIN | HINA | HINB |
| LPS *E. coli* O111:B4 (LPS-EB) | 4.2 | 116.5 | 85.3 | 3.3 |
| LPS *E. coli* F583 (LPS-F583) | 4.3 | - | - | - |
| LPS *P. gingivalis* (LPS-PG) | 12.0 | 46.9 | 84.0 | 1.6 |
| LPS *R. sphaeroides* (LPS-RS) | 19.3 | - | - | - |
| DPLA *E. coli* F583 (DPLA) | 0.9 | 47.5 | 67.7 | 2.8 |
| MPLA *E. coli* F583 (MPLA) | 1.2 | 47.2 | 94.1 | 2.7 |
| Detoxified LPS *E. coli* O111:B4 (detoxLPS) | > 20 | - | - | - |

$K_D$, equilibrium dissociation constant

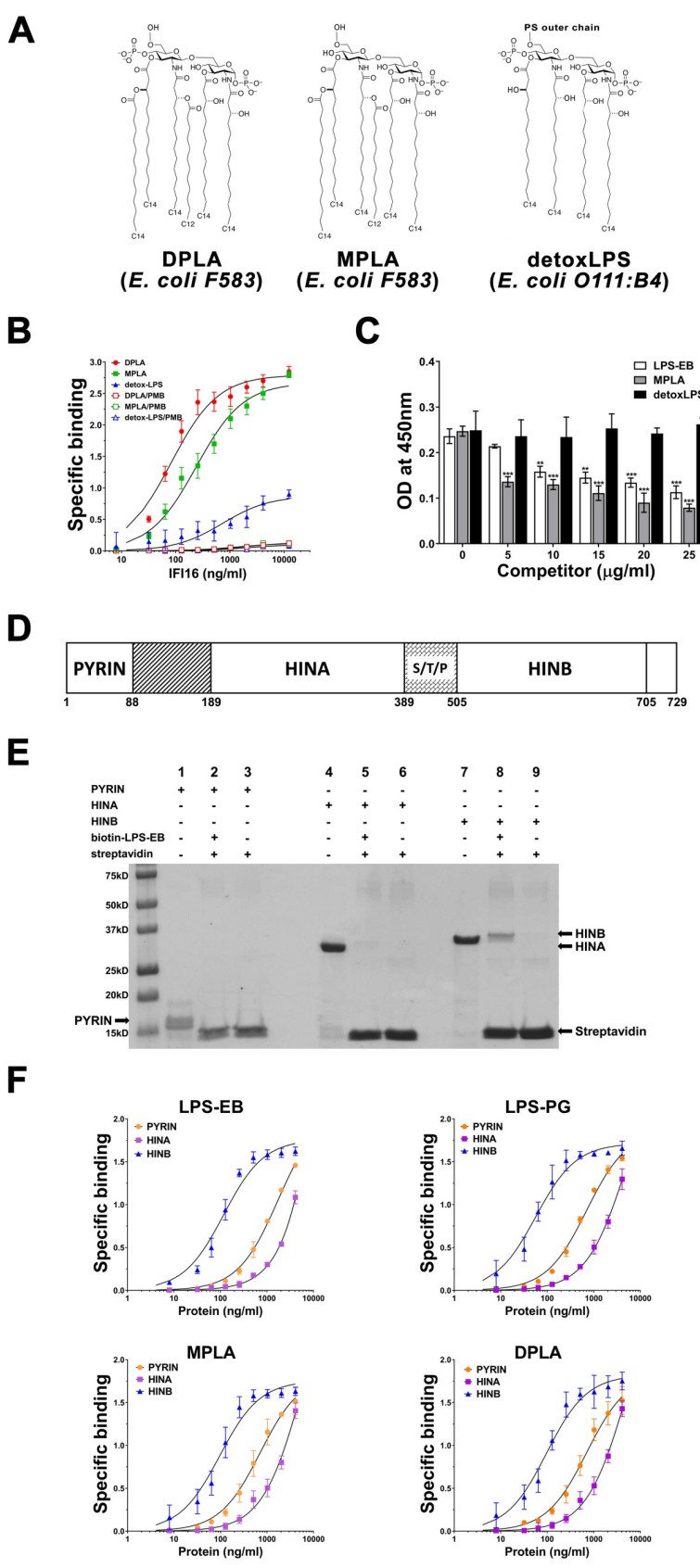

**Fig 2. FI16 binds to the lipid A moiety of LPS through its HINB domain. (A)** Structures of di- or mono-phosphorylated lipid A from *E. coli* F583 (DPLA and MPLA, respectively) and detoxified LPS (detox-LPS) derived from *E. coli* O111:B4. PS-outer chain = polysaccharide outer chain. **(B)** Saturation binding experiments with increasing amount of recombinant IFI16 (from 8 to 12,288 ng/ml) and 10 μg/ml of MPLA (green line), DPLA (red line) or detox-LPS (purple line). Binding was detected by ELISA as described in the legend to Fig 1G. Data are expressed as mean values ± SD of three different experiments. **(C)** Competition ELISA assay for LPS-EB binding to IFI16 with increasing amount of LPS-EB, MPLA or detox-LPS as competitors. Briefly, microtiter plates were coated with 1 μg/ml LPS-EB, then 2 μg/ml of IFI16 were added to the wells in the presence of increasing concentration (5 to 25 μg/ml) of competitor. Binding was detected by ELISA as described in the legend to Fig 1F. Data are expressed as mean values ± SD of three independent experiments (***P<0.001, **P<0.01, Student's t test). **(D)** Domain organization of the IFI16 protein. The numbers represent the amino acid positions based on NCBI Reference Sequence NP_005522. From the N- to the C-terminal (left to right), IFI16 comprises a pyrin domain involved in protein-protein interaction, and two hematopoietic interferon-inducible nuclear protein with 200-amino-acid repeats (HINA and HINB) domains, which are a hallmark of the absent in melanoma 2-like receptors (ALRs). S/T/P = serine/threonine/proline-rich repeats, which are regulated by alternative mRNA splicing. **(E)** Coomassie brilliant blue staining of pull-down assays performed with 3 μg of recombinant PYRIN, HINA, and HINB domains, in presence or absence of biotin-labeled LPS from *E. coli* O111:B4 (biotin-LPS-EB). **(F)** Saturation binding experiments performed by using increasing amount (8 to 4,096 ng/ml) of recombinant PYRIN, HINA or HINB domains (orange, purple and blue lines, respectively) and 10 μg/ml of LPS-EB, LPS-PG, MPLA or DPLA. Anti-IFI16 antibodies against the N- or C-terminus of the protein and HRP-labeled anti-rabbit IgG were added as primary and secondary antibody, respectively, and binding detected in ELISA at 450 nm. Data are expressed as mean values ± SD of three independent experiments.

plates followed by the addition of a mixture of a constant amount of IFI16 and increasing concentrations of LPS-EB, MPLA or detoxLPS, in this case used as competitors. As expected, addition of LPS-EB reduced IFI16 binding to immobilized LPS in a concentration-dependent manner (Fig 2C, white bars). Interestingly, a concentration of 5 μg/ml of MPLA was sufficient enough to achieve a much stronger reduction in IFI16 binding to LPS compared to a similar dose of LPS-EB (Fig 2C, grey bars). Binding inhibition was further enhanced at higher concentrations of MPLA, but the difference between the two variants was less evident. By contrast, detoxLPS did not interfere with the binding of IFI16 to the canonical agonist LPS, even at the highest concentrations used (25 μg/ml: LPS-*EB vs*. detoxLPS, *P* = 0.0059; MPLA *vs*. detoxLPS, *P* < 0.0004; LPS-EB *vs*. MPLA, ns; unpaired t-test) (Fig 2C, black bars). Taken together, these findings indicate that lipid A is the LPS moiety involved in the binding to IFI16 and that the heteropolysaccharide outer chain, absent in MPLA, might constitute a steric hindrance for such interaction.

To identify the domain of IFI16 mediating binding to LPS, an *in vitro* pull-down assay was performed using three distinct recombinant domains of IFI16 spanning either the N-terminal portion containing the pyrin domain (PYRIN) or each of the 200 amino acid-long HIN domains (namely HINA or HINB) (Fig 2D). As shown in Fig 2E, a signal at ~35 kDa was only detected when the HINB fragment was incubated with biotin-LPS-EB bound to streptavidin beads (lane 9), while neither the PYRIN nor the HINA fragment was co-precipitated in the presence of biotinylated LPS (lanes 3 and 6). To corroborate these data, an *in vitro* pull-down assay was performed using a truncated variant of IFI16 lacking the HINB domain (IFI16Δ-HINB) (S1A Fig). As expected, no binding was observed when IFI16ΔHINB was incubated with biotin-LPS-EB and streptavidin beads (S1B Fig, lane 2), confirming that the HINB is required for LPS binding. To further support a role of the HINB domain in mediating the binding of IFI16 to LPS, we performed saturation binding experiments using increasing concentrations of the three IFI16 domains (*i.e*., PYRIN, HINA, and HINB) with fixed amounts of different interactors. As shown in Fig 2F (blue lines), the HINB domain was able to bind to both LPS variants, displaying either strong or weak TLR4 agonist activity, as well as to lipid A. The binding was not affected by the origin of bacterial LPS or by the number of phosphate groups, and displayed $K_D$ values in a similar range to that obtained with the full-length recombinant IFI16 protein (Table 1). Conversely, the PYRIN (orange lines) and HINA (purple lines)

domains displayed very low affinity for the immobilized molecules when compared to HINB, with $K_D$ values indicative of unspecific binding (Table 1). Thus, the HINB domain displays the highest affinity for LPS, indicating that HINB may play a major role in the interaction between IFI16 and LPS.

## Only potent TLR4-activating endotoxins can potentiate the proinflammatory activity of IFI16

The results so far obtained prompted us to investigate whether IFI16 binding to the strong agonist variant LPS-EB would modulate IFI16-mediated transcriptional activation of proinflammatory cytokines *in vitro*. For these experiments, in addition to the standard human monocytic cell line THP-1, we chose as a model the renal tubular carcinoma cell line 786-O. We first assessed protein expression levels of the main components of the LPS recognition complex (*i.e.*, TLR4, MD2, MyD88 and CD14) by Western blotting and/or flow cytometry (S2 Fig). While TLR4 and MD2 were expressed at similar levels in both cell lines, CD14 expression was 4-fold lower in THP-1 *vs.* 786-O cells, as judged by FACS analysis (S2B Fig), in good agreement with a previous report [32]. The expression of the TLR4 canonical adaptor MyD88 was similar in both cell lines (S2A Fig).

Next, cells were stimulated with full-length IFI16 or the IFI16ΔHINB variant, alone or preincubated with LPS-EB, and then total RNA was extracted to assess mRNA levels of a panel of proinflammatory cytokines (Fig 3A). Consistent with IFI16 acting as a DAMP, IL-6, IL-8 and tumor necrosis factor-α (TNF-α) mRNA levels were strongly upregulated in cells treated with IFI16 alone when compared to mock- or LPS-treated cells in the presence of either low or high LPS concentration. Interestingly, we detected a further increase in mRNA expression levels of the aforementioned genes in cells treated with the IFI16/LPS-EB complex compared to IFI16 alone—*i.e.*, 1.8- and 1.6-fold induction for IL-6; 1.7- and 1.3-fold induction for IL-8; and 2.1- and 1.6-fold induction for TNF-α in 786-O and THP-1 cells, respectively. Likewise, IL-1β gene expression levels were also significantly induced by IFI16 alone or IFI16/LPS-EB complex treatment of THP-1 cells—*i.e.*, 72- and 83-fold induction, respectively—and, albeit to a lower extent, 786-O cells—*i.e.*, 13- and 27-fold induction, respectively. When IFI16ΔHINB alone or pre-incubated with LPS-EB was used to stimulate the cells, the degree of cytokine induction was similar to that observed with the full-length protein, while it was not enhanced following pre-incubation with LPS-EB.

Altogether, these findings further strengthen the notion that the HINB moiety is necessary for the formation of the functional IFI16/LPS complex.

Consistent with the mRNA data, the amounts of IL-6, IL-8 and TNF-α secreted into the culture supernatants were significantly higher in cells treated with the IFI16/LPS-EB complex than those of cells treated with IFI16 alone, while they did not vary upon pre-incubation with LPS-EB in the case of the IFI16ΔHINB variant (Fig 3B). In contrast, neither IFI16 nor IFI16Δ-HINB *per se* or after forming a complex with LPS induced IL-1β release in both cell lines, indicating lack of inflammasome-mediated IL-1β processing at 24 h post-treatment.

Next, we asked whether the LPS derivatives DPLA and MPLA or the TLR4 antagonist LPS-RS would be equally able to modulate the biological activity of IFI16. The full agonist LPS-F583, from which DPLA and MPLA were derived, was included as positive control (full TLR4 activator). Cells were treated with the aforementioned compounds, alone or pre-complexed with IFI16 protein, and then total RNA and supernatants were collected to assess the mRNA expression and cytokine production profiles of IL-6, IL-8 and TNF-α. As expected, cells treated with the IFI16/LPS-F583 displayed a similar transcriptional activation pattern to that previously observed in IFI16/LPS-EB-treated cells (Fig 4A). On the other hand, when cells

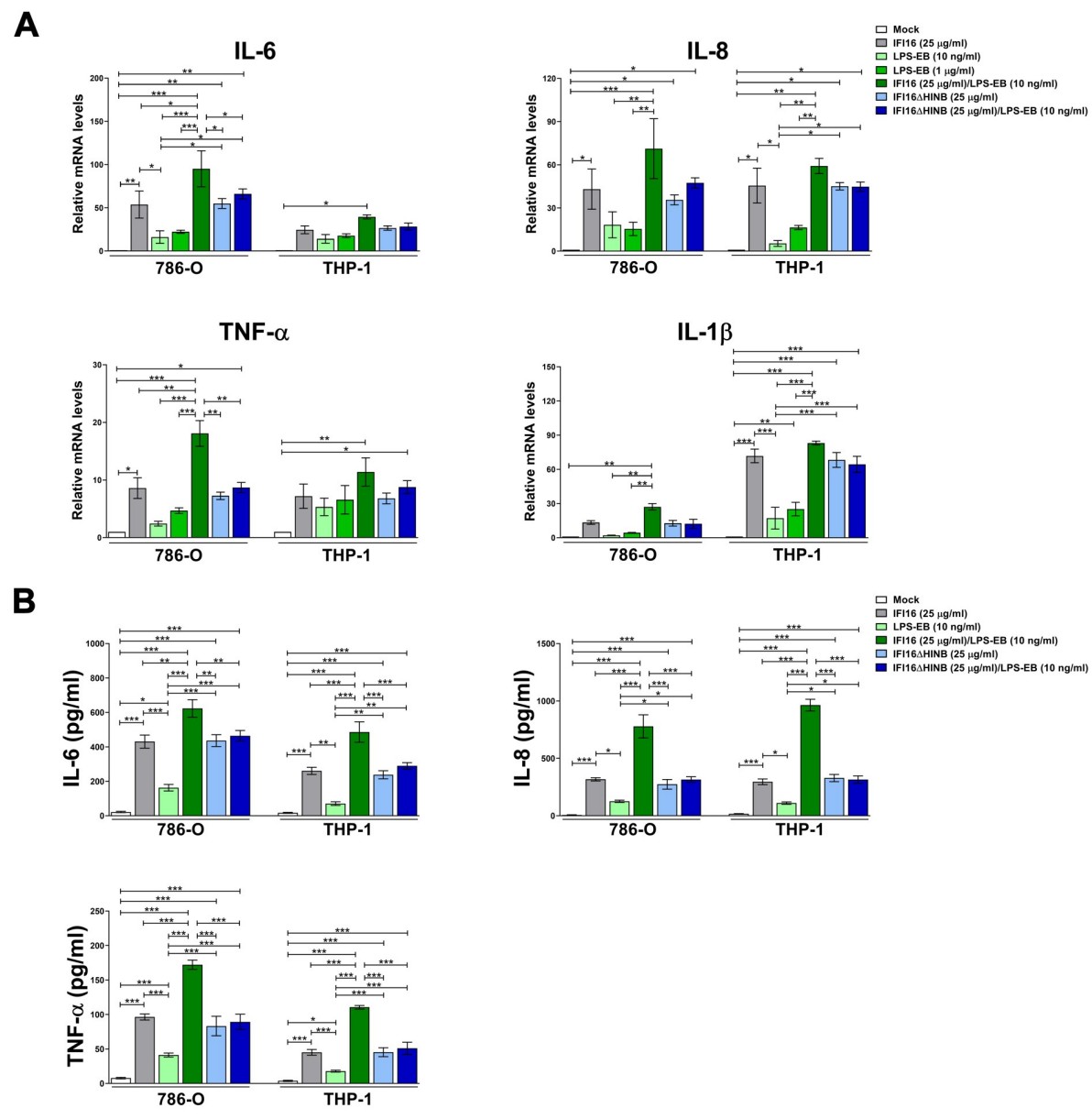

**Fig 3. IFI16 proinflammatory activity is potentiated by the strong TLR4 activator LPS-EB. (A)** qRT-PCR analysis of IL-6, IL-8, TNF-α and IL-1β mRNA expression levels in 786-O or THP-1 cells stimulated for 24 h with IFI16 (25 μg/ml), IFI16ΔHINB (25 μg/ml), LPS from *E. coli* O111:B4 (LPS-EB, 10 ng/ml or 1 μg/ml), IFI16/LPS-EB complex (preincubated O/N at 4˚C), IFI16ΔHINB/LPS-EB (preincubated O/N at 4˚C), or left untreated (mock). Values are normalized to GAPDH mRNA and plotted as fold induction over mock-treated cells. qRT-PCR data are presented as mean values of biological triplicates. Error bars indicate SD (*$P < 0.05$, **$P < 0.01$, ***$P < 0.001$; two-way ANOVA followed by Dunnett's test). **(B)** Protein concentration of IL-6, IL-8 and TNF-α evaluated by ELISA in supernatants derived from 786-O or THP-1 cells stimulated for 24 h as described in **A**. Data are expressed as mean values ± SD of three independent experiments (*$P < 0.05$, **$P < 0.01$, ***$P < 0.001$; two-way ANOVA followed by Dunnett's test).

were treated with IFI16 complexed with LPS-RS, MPLA or DPLA, we failed to observe any transcriptional enhancement in comparison with IFI16 alone. A similar pattern was found when the same cytokines were measured in the culture supernatants by ELISA (Fig 4B), although a significant increase in IFI16-induced secretion of IL-6 and IL-8 was only observed when cells were treated with the IFI16-DPLA complex—*i.e.*, 1.4-fold induction for IL-6 in THP-1 cells, 1.3- and 1.8-fold induction for IL-8 in 786-O and THP-1, respectively.

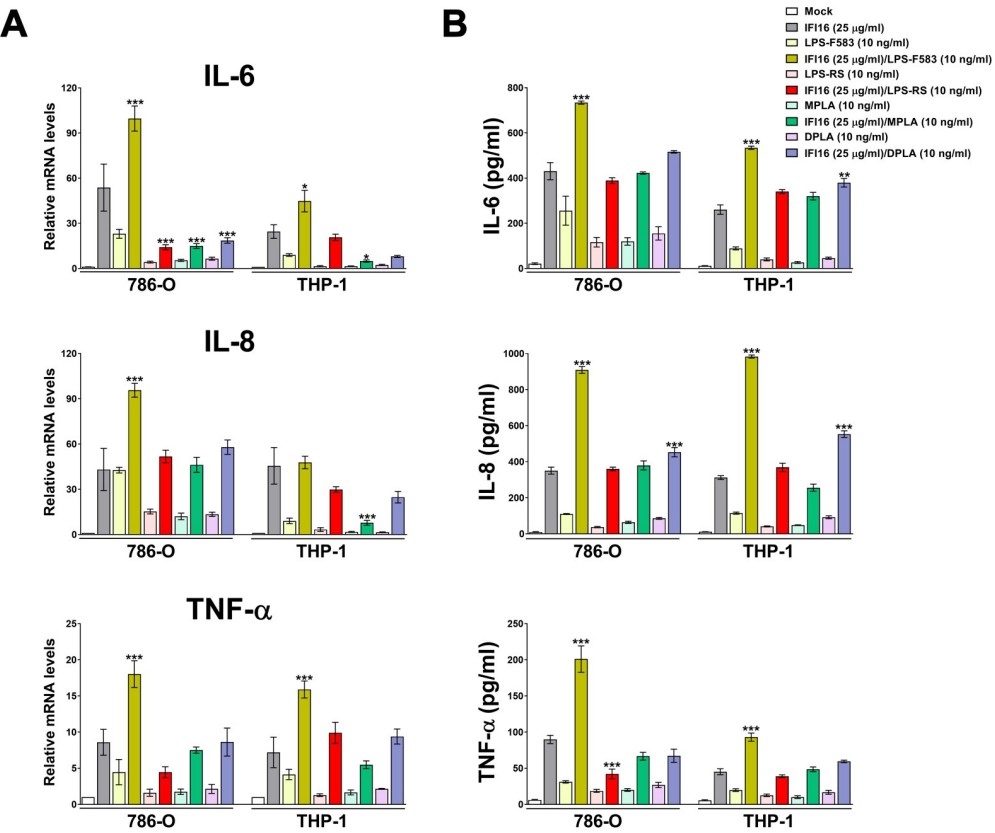

**Fig 4. Weak TLR4-activating LPS variants and the TLR4 antagonist LPS-RS do not potentiate IFI16 proinflammatory activity. (A)** qRT-PCR analysis of IL-6, IL-8 and TNF-α mRNA expression levels in 786-O or THP-1 cells stimulated for 24 h with or without IFI16 (25 μg/ml), LPS from *E. coli* F583 (LPS-F583, 10 ng/ml) or LPS from *R. sphaeroides* (LPS-RS, 10 ng/ml), MPLA (10 ng/ml), DPLA (10 ng/ml) or in the presence of one of the following complexes: IFI16/LPS-F583, IFI16/LPS-RS, IFI16/MPLA or IFI16/DPLA. Values are normalized to GAPDH mRNA and plotted as fold induction over mock-treated cells. qRT-PCR data are presented as mean values of biological triplicates. Error bars indicate SD, and the *P* values refer to comparisons between IFI16 *vs*. IFI16/LPS or IFI16/lipid A complex-treated cells (*$P < 0.05$, **$P < 0.01$, ***$P < 0.001$; two-way ANOVA followed by Dunnett's test). **(B)** Protein concentration of IL-6, IL-8 and TNF-α evaluated by ELISA in supernatants derived from 786-O or THP-1 cells stimulated for 24 h as described in the legend to panel **A**. Data are expressed as mean values ± SD of three independent experiments (*$P < 0.05$, **$P < 0.01$, ***$P < 0.001$; ns, not significant; two-way ANOVA followed by Dunnett's test). The *P* values are relative to comparisons between IFI16- and IFI16/LPS- or IFI16/lipid A-treated cells.

Collectively, these data indicate that the proinflammatory activity of full-length IFI16 is potentiated when this protein is complexed with potent TLR4-activating LPS variants *via* the HINB moiety, while it is not affected when it forms a complex with the TLR4 antagonist LPS-RS or the weak agonists MPLA and DPLA.

## IFI16 exerts its proinflammatory activity in a TLR4/MyD88-dependent fashion

Since we had previously implicated TLR4 signaling in IFI16-mediated cytokine release in endothelial cells [24], we sought to determine whether ablation of the TLR4/MD2 complex would affect IFI16/LPS proinflammatory activity. To this end, we performed gene silencing of TLR4 and MD2 genes in both 786-O and THP-1 cells, achieving complete knockdown of both genes, as judged by immunoblotting and flow cytometric analysis (S3A–S3C Fig). As shown in S4 Fig, the transcriptional activation of IL-6, IL-8 and TNF-α genes (panels A and B) as well as

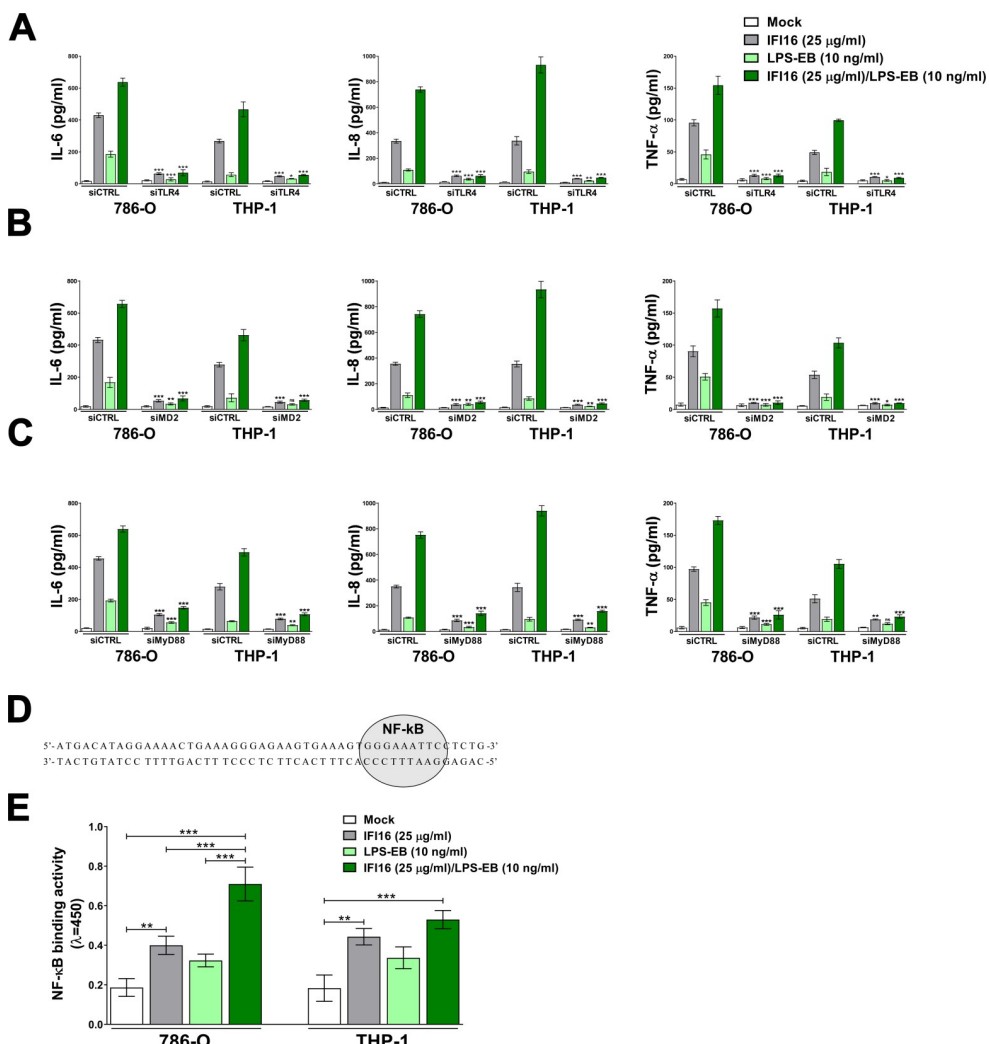

**Fig 5. IFI16 exerts its proinflammatory activity in a TLR4/MyD88-dependent fashion. (A-C)** Protein concentration of IL-6, IL-8 and TNF-α measured by ELISA in supernatants derived from 786-O or THP-1 cells transfected for 48 h with scramble control (siCTRL), or siRNAs against TLR4 (siTLR4, **A**), MD2 (siMD2, **B**) or MyD88 (siMyD88, **C**). Cells were then stimulated for 24 h with IFI16 (25 μg/ml), LPS from *E. coli* O111:B4 (LPS-EB, 10 ng/ml) or IFI16/LPS-EB complex (preincubated O/N at 4°C), or left untreated (mock). Data are expressed as mean values ± SD of three independent experiments (*$P < 0.05$, **$P < 0.01$, ***$P < 0.001$, ns: not significant; unpaired Student's *t*-test for comparison of silenced cells *vs.* their relative control counterpart). **(D)** Schematic representation of the probe containing the NF-κB binding site (highlighted in grey). **(E)** 786-O cells or THP-1 cells were stimulated with IFI16 (25 μg/ml), LPS from *E. coli* O111:B4 (LPS-EB, 10 ng/ml) or IFI16/LPS-EB complex (preincubated O/N at 4°C), or left untreated (mock). After 2 h, the cells were lysed and the nuclear fraction was analyzed for NF-κB binding activity using the Universal EZ-TFA transcription factor assay colorimetric kit and the probe described in **D**. Data are expressed as mean values ± SD of three independent experiments (*$P < 0.05$, **$P < 0.01$, ***$P < 0.001$; two-way ANOVA followed by Dunnett's test).

the release of the same cytokines (Fig 5A, B) in the culture supernatants upon exposure to both IFI16 or IFI16/LPS-EB was almost completely abolished in siTLR4- and siMD2-silenced cells when compared to siRNA control (siCTRL)-transfected cells. Similar results were obtained in both 786-O and THP-1 cells, indicating that IFI16 signaling through the TLR4-MD2 complex is not cell type-specific.

Upon LPS stimulation, TLR4 induces two independent signaling pathways regulated by either the TIRAP/MyD88 or TRAM/TRIF pair of adaptors, which promote the production of

proinflammatory cytokines and type I interferons (IFN-I), respectively [33,34]. As IFN-β could never be detected in IFI16- or IFI16/LPS-stimulated cells, we assumed that the TRAM-TRIF pathway would not play a role in our model. To address a potential role of the TIRAP-MyD88 complex, we performed siRNA-mediated knockdown of MyD88 in both 786-O and THP-1 cells (S3D Fig). In MyD88-silenced cells treated with IFI16 alone or IFI16/LPS-EB, transcription of IL-6, IL-8 and TNF-α genes (S4C Fig) and release of the corresponding cytokines (Fig 5C) were dramatically reduced in comparison with siCTRL-transfected cells, indicating that IFI16 or the IFI16/LPS complex signals through the TIRAP-MyD88 axis. Fittingly, ELISA-based transcription factor binding assay, performed using a probe containing the NF-κB binding site (Fig 5D), showed NF-κB binding activity to be significantly increased in cells challenged with IFI16 alone or pre-complexed with LPS-EB in comparison with untreated cells—*i.e.*, 2.2- and 3.9-fold induction in 786-O cells; 2.4- and 2.9-fold induction in THP-1 cells, respectively (Fig 5E). Overall, these results demonstrate that IFI16-mediated proinflammatory cytokine production requires the TLR4-MD2/TIRAP-MyD88 signaling pathway, which then promotes NF-κB nuclear binding activity to target DNA.

Finally, to circumvent potential issues of structural or functional differences between mammalian or bacterial expressed IFI16, we used wild-type and IFI16-knockout (IFI16$^{-/-}$) human osteosarcoma U2OS cells as a source of endogenous IFI16 released under stress stimuli. As shown in Fig 6A, and consistent with our previous report [8], UVB treatment (800 Jm$^{-2}$ for 16h) led to massive release of IFI16 in the culture supernatants of U2OS cells that, as expected, did not occur in their IFI16$^{-/-}$ counterparts, thus serving as IFI16-depleted supernatant. The resulting conditioned media were preincubated with or without LPS-EB or LPS-RS and then added to THP-1 cells. After 24 h, the supernatants of the THP-1 cultures were harvested and assessed for cytokine expression by ELISA. Consistent with the results obtained with recombinant IFI16, exposure of THP-1 cells to conditioned medium from UVB-treated U2OS cells significantly stimulated the release of IL-6, IL-8 and TNF-α by THP-1 cells when compared to mock-treated cells—*i.e.*, 20.9-fold higher for IL-6; 83-fold for IL-8; and 33.4-fold for TNF-α (Fig 6B). When THP-1 cells were pre-treated with anti-TLR4 antibodies, the stimulatory activity of the conditioned medium form UVB-treated U2OS cells dropped significantly. Notably, cytokine release was significantly lower in THP-1 cells treated with conditioned medium from UVB-treated U2OS-IFI16$^{-/-}$ cells when compared to their UVB-treated normal counterparts—*i.e.*, 2.8-fold lower for IL-6; 2.7-fold for IL-8; and 2.6-fold for TNF-α—, indicating that the effects observed were specifically due to the secretion of IFI16 protein. Consistent with the data obtained with the recombinant protein, cytokine release was enhanced when the conditioned media were preincubated with LPS-EB and, to a higher extent, with the conditioned medium of UVB-treated U2OS cells when compared to that of UVB-treated U2OS-IFI16$^{-/-}$ cells. As expected, this enhancement was not observed when LPS-RS was used.

## IFI16 binds to TLR4 *in vivo* and *in vitro*

We next sought to determine whether IFI16 could also bind to the TLR4/MD2 complex *in vivo*. To this end, co-immunoprecipitation assays were performed where TLR4 and interacting partners were immunoprecipitated using an anti-TLR4 antibody pre-adsorbed on protein G beads. The resulting immune complexes were then analyzed by SDS-PAGE followed by immunoblotting for TLR4, MD2, and IFI16. As shown in Fig 7A, IFI16 co-immunoprecipitated with TLR4/MD2 receptor when total extracts from cells treated with either IFI16 alone or IFI16/LPS-EB complex were used (lane 2 and 4, respectively). The specificity of this interaction was attested by the absence of co-immunoprecipitated IFI16 in extracts from cells untreated or treated with LPS-EB alone (lane 1 and 3, respectively). To ensure that residual DNA potentially

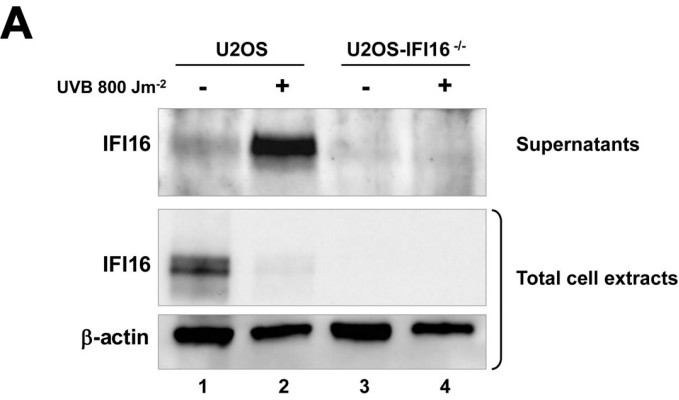

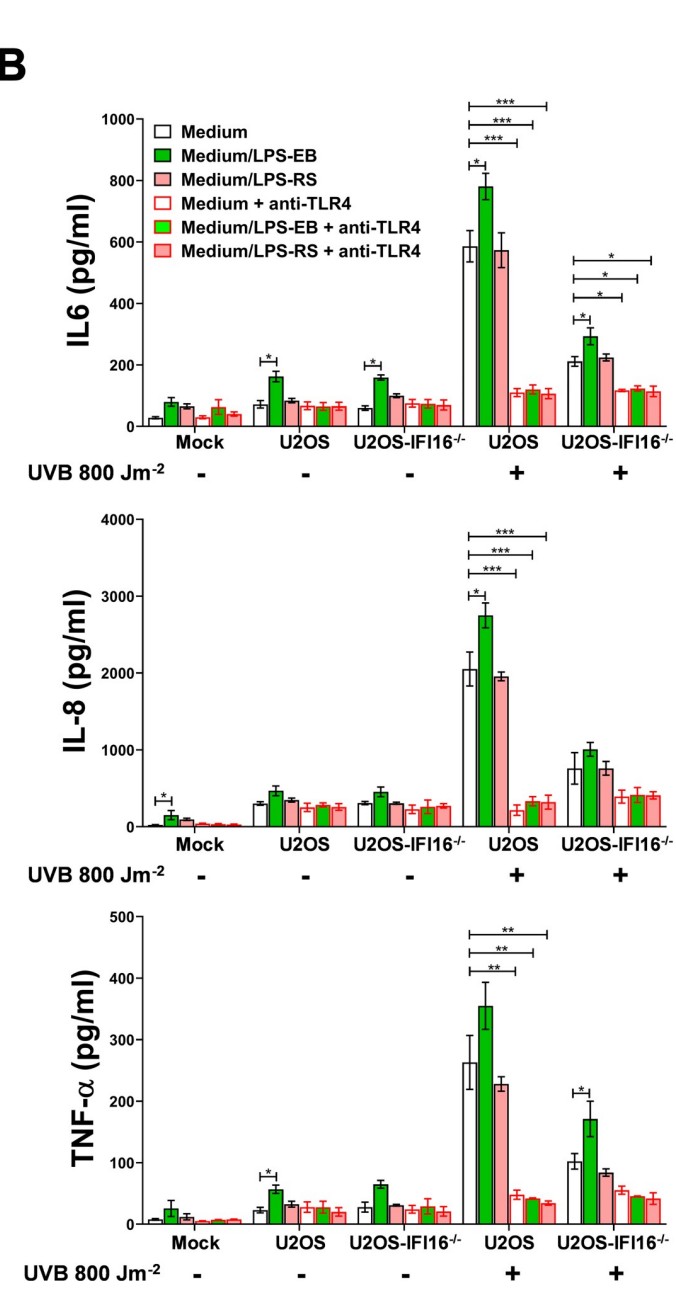

**Fig 6. Endogenous IFI16 is released by UVB-exposed U2OS cells and triggers proinflammatory cytokines production in a TLR4-dependent fashion. (A)** Western blot analysis of IFI16 in culture supernatants and total cell extracts of UVB-exposed (0 or 800 Jm$^{-2}$) U2OS or U2OS-IFI16$^{-/-}$ cells at 16 h after treatment. β-actin cellular expression was used for protein loading control. Data are representative of three independent experiments with similar results. **(B)** Protein concentration of IL-6, IL-8 and TNF-α evaluated by ELISA in supernatants derived from THP-1 cells stimulated for 24 h in the presence or absence of anti-TLR4 neutralizing antibodies (10 μg/ml) using conditioned medium from UVB-exposed (0 or 800 Jm$^{-2}$) U2OS and U2OS-IFI16$^{-/-}$ cells, or complete medium (mock), preincubated (O/N at 4˚C), or not, with LPS from *E. coli* O111:B4 (LPS-EB, 10 ng/ml), or LPS from *R. sphaeroides* (LPS-RS, 10 ng/ml). Values were normalized to the initial protein concentration of the analyzed cytokines in the supernatants used for the treatment. Data are expressed as mean values ± SD of three independent experiments. The *P* values refer to comparison in each group with cells treated only with the medium without any addiction (white bar and black border; $^{*}P < 0.05$, $^{**}P < 0.01$, $^{***}P < 0.001$; two-way ANOVA followed by Dunnett's test).

present in the protein extracts would not affect Co-IPs, whole-cell extracts were treated with DNase and then subjected to Co-IP. As shown in Fig 7A (lane 5 and 6, respectively), the interaction between IFI16 or IFI16/LPS and TLR4 was maintained also in protein extracts obtained from DNase-treated cells, indicating that the interaction between these molecules is not mediated by DNA binding.

The specificity of the interaction between IFI16 and TLR4 was then evaluated by surface plasmon resonance (SPR). Briefly, recombinant TLR4 was directly immobilized on a CM5 sensor chip by amine coupling and then probed with increasing concentration of recombinant IFI16—from 31.25 nM to 1 μM. As shown in Fig 7B, the resulting SPR sensorgrams revealed significant binding between TLR4 and IFI16 with an equilibrium dissociation constant (K$_D$) of 0.13 μM and a kinetic profile typical of dynamically interacting partners, with the dissociation rate being compatible with a rapid stimulation turnover of the ligand (*i.e.*, IFI16) on the TLR4 receptor. Thus, taken together these results indicate that the proinflammatory activity of IFI16, either alone or pre-complexed with LPS, is mediated by the TLR4/MD2/MyD88/NF-κB signaling pathways and requires a direct interaction between IFI16 and TLR4.

## The IFI16/LPS complex proinflammatory activity is not affected by the presence of free LPS

To gain more insights into the biological relevance of the IFI16/LPS complex *vs.* LPS, we sought to determine the proinflammatory activity of IFI16 or IFI16/LPS complex in the presence or absence of equal amounts of LPS simultaneously added to the cells. For this purpose, 786-O and THP-1 cells were stimulated with an array of different combinations as indicated in Fig 8. When IFI16 and LPS-EB were simultaneously added to the cells, without any pre-incubation step, induction of IL-6, IL-8 and TNF-α at both the mRNA and protein levels (Fig 8A and 8B, respectively) was comparable to that observed using LPS-EB alone, indicating that the affinity of IFI16 for the TLR4/MD2 receptor is lower than that of LPS. In contrast, when LPS-EB was simultaneously added to the cells together with the pre-formed IFI16/LPS-EB complex, induction of IL-6, IL-8 and TNF-α at both the mRNA and protein levels was comparable or even higher than that observed in cells stimulated with the IFI16/LPS-EB complex alone, suggesting that the affinity of the IFI16/LPS-EB complex for the TLR4 receptor is stronger than that of LPS-EB alone. Similar results were obtained when LPS-RS, a TLR4 antagonist, was used with the same combination treatment. Again, IFI16/LPS-RS complex activity, as measured by the induction of IL-6, IL-8 and TNF-α, was not affected by simultaneous addition of equal amounts of LPS-RS. Taken together, these findings clearly show that once IFI16 is complexed with LPS its proinflammatory activity is not affected by the simultaneous addition of LPS, regardless of its bacterial origin.

To better clarify the dynamics of interaction between IFI16/LPS-EB complex and TLR4/ MD2 receptor, we performed further SPR analyses by means of a CM5 sensor chip coated with

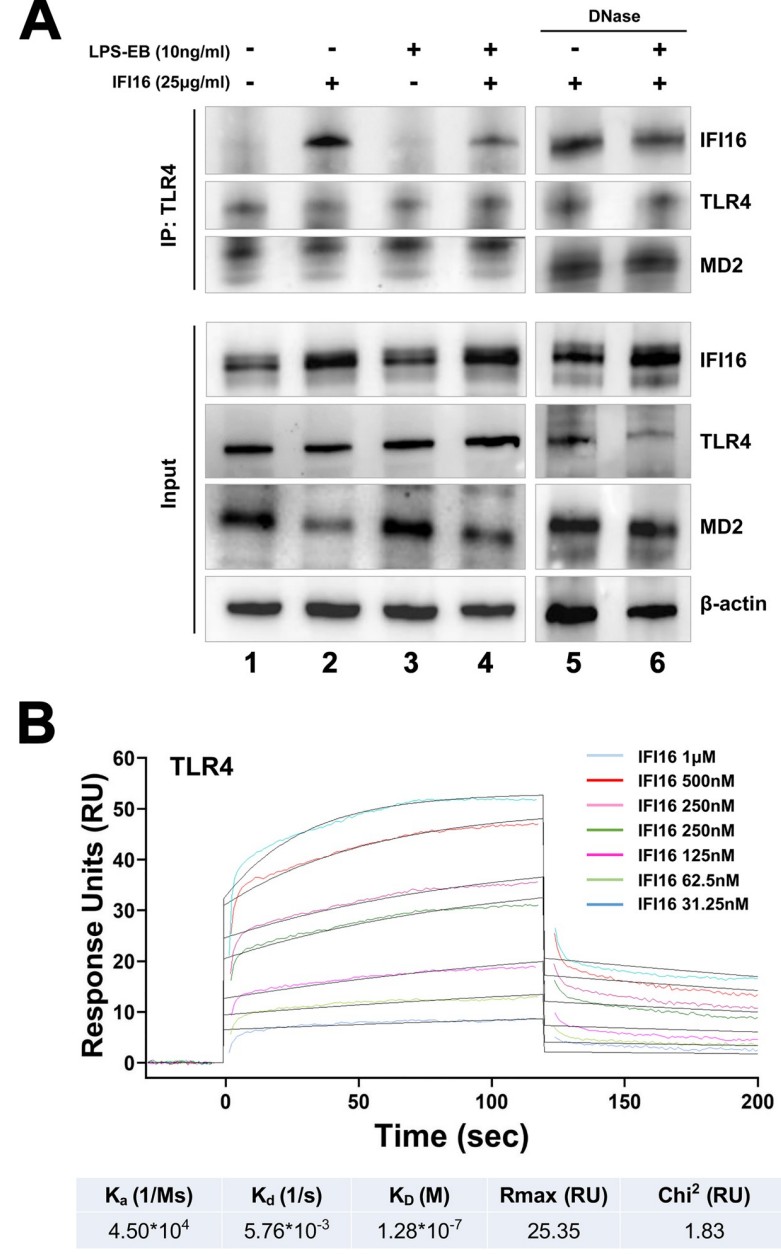

**Fig 7. IFI16 binds to TLR4 *in vitro* and *in vivo*. (A)** 786-O cells were stimulated for 1 h in the presence or absence of the indicated concentrations of IFI16, LPS from *E. coli* O111:B4 (LPS-EB), or IFI16/LPS-EB complex. Total cell extracts, untreated or DNase I-treated, were subjected to immunoprecipitation using a TLR4 monoclonal antibody. Immunoprecipitates and whole-cell lysates were analyzed by immunoblotting with anti-IFI16, anti-TLR4 or anti-MD2 antibodies. β-actin protein expression was used for protein loading control. Data are representative of three independent experiments with similar results. **(B)** Surface plasmon resonance (SPR) analyses of IFI16 binding to immobilized TLR4. After immobilization of TLR4 on the CM5 sensor chip surface, increasing concentration of IFI16 (31.25–1,000 nM) diluted in running buffer were injected over immobilized TLR4. IFI16 binds to TLR4 with an equilibrium dissociation constant ($K_D$) of 0.13 μM. Data are representative of three independent experiments.

recombinant TLR4/MD2. As shown in Table 2, the SPR-based studies demonstrated that the three receptor partners (*i.e*., IFI16, LPS-EB and the IFI16/LPS-EB complex) had association rate ($K_a$) values in the same range, with IFI16/LPS-EB showing a 3- and 2-fold faster association compared to IFI16 or LPS-EB alone, respectively. Interestingly, in agreement with our

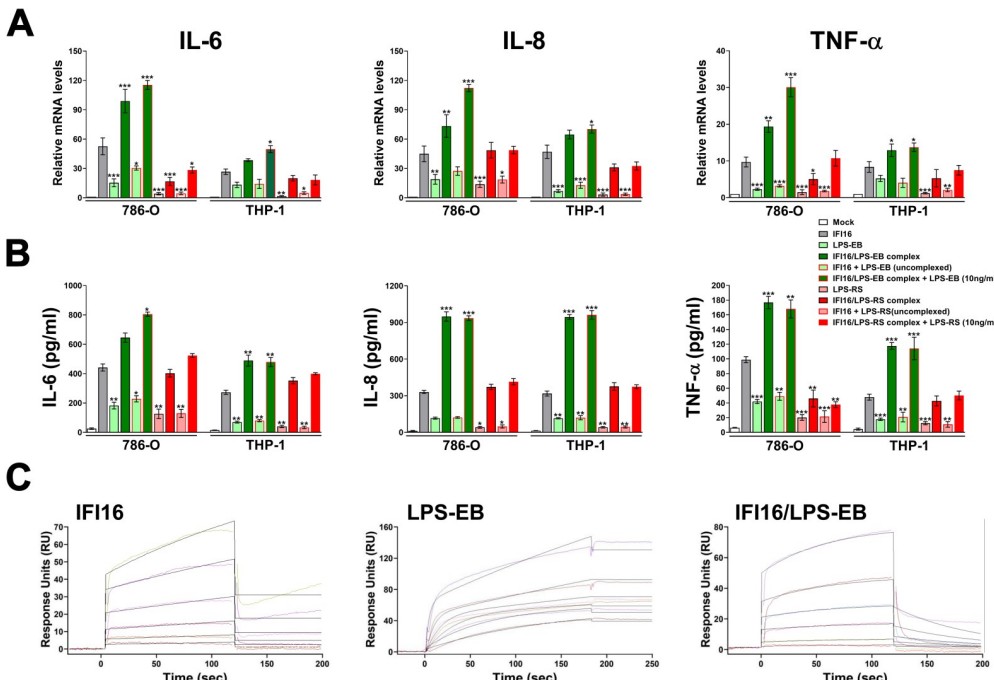

**Fig 8. The IFI16/LPS complex proinflammatory activity is not affected by the presence of free LPS. (A)** qRT-PCR analysis of IL-6, IL-8, TNF-α and mRNA expression levels in 786-O or THP-1 cells stimulated for 24 h with IFI16 (25 μg/ml), LPS from *E. coli* O111:B4 (LPS-EB, 10 ng/ml), IFI16/LPS-EB complex, IFI16 + LPS-EB (not complexed), IFI16/LPS-EB complex + LPS-EB (10 ng/ml), LPS from *Rhodobacter sphaeroides* (LPS-RS, 10 ng/ml), IFI16/LPS-RS complex, IFI16 + LPS-RS (not complexed), IFI16/LPS-RS complex + LPS-RS (10 ng/ml), or left untreated (mock). Values were normalized to GAPDH mRNA and plotted as fold induction over mock-treated cells. The *P* values refer to comparisons with IFI16-treated cells (*$P < 0.05$, **$P < 0.01$, ***$P < 0.001$; one-way ANOVA followed by Dunnett's test). **(B)** Protein concentration of IL-6, IL-8 and TNF-α was measured by ELISA in supernatants derived from 786-O or THP-1 cells stimulated for 24 h as described in **A**. Data are expressed as mean values ± SD of three independent experiments. The *P* values refer to comparisons with IFI16-treated cells (*$P < 0.05$, **$P < 0.01$, ***$P < 0.001$; one-way ANOVA followed by Dunnett's test). **(C)** Surface plasmon resonance (SPR) analyses of IFI16, LPS-EB and IFI16/LPS-EB complex binding to immobilized TLR4/MD2 receptor. After immobilization of recombinant TLR4/MD2 on the CM5 sensor chip surface, increasing concentration of the different analytes—31.25–1000 nM for IFI16, 3.125–100 μM for LPS-EB, 31.25–1000 nM for IFI16/LPS-EB complex—diluted in running buffer were injected over immobilized TLR4/MD2. IFI16, LPS-EB, and IFI16/LPS-EB bind to TLR4/MD2 with an equilibrium dissociation constant (K_D) of 0.68 μM, 0.15 nM, and 4.01 μM, respectively. Data are representative of three independent experiments.

previous findings, IFI16 interacted with the TLR4/MD2 complex in a concentration-dependent manner with a $K_D$ of 0.68 μM, as calculated by the evaluation of the sensorgrams in Fig 8C. Moreover, LPS-EB alone revealed a much higher affinity, with a $K_D$ of 0.15 nM. In contrast, IFI16/LPS-EB complex bound to TLR4/MD2 with 6-fold lower affinity ($K_D = 4.01$ μM)

**Table 2. Binding kinetics of IFI16, LPS-EB and IFI16/LPS-EB to TLR4/MD2 receptor.**

|  | $K_a$ (1/Ms) | $K_d$ (1/s) | $K_D$ (M) | Rmax (RU) | Chi$^2$ (RU) |
|---|---|---|---|---|---|
| **IFI16** | $9.07 \times 10^3$ | $6.14 \times 10^{-3}$ | $6.77 \times 10^{-7}$ | 26.09 | 1.83 |
| **LPS-EB** | $6.84 \times 10^3$ | $1.00 \times 10^{-6}$ | $1.47 \times 10^{-10}$ | 745.00 | 14.30 |
| **IFI16/LPS-EB** | $3.57 \times 10^3$ | $1.43 \times 10^{-2}$ | $4.01 \times 10^{-6}$ | 90.30 | 6.89 |

$K_a$, association rate constant; M, molarity; s, seconds; $K_d$, dissociation rate constant; $K_D$, equilibrium dissociation constant; Rmax, maximum response; RU, response units; Chi$^2$, average squared residual.

when compared to IFI16 alone. The lower $K_D$ displayed by LPS-EB was mainly due to the contribution of a lower dissociation rate constant ($K_d = 1.00^*10^{-6}$ s$^{-1}$), indicating that the kinetics of LPS-EB dissociation from the receptor is very slow. Interestingly, the sensorgrams demonstrated a much higher dissociation rate for both free IFI16 ($K_d = 6.14^*10^{-3}$ s$^{-1}$) and the IFI16/LPS-EB complex ($K_d = 1.43^*10^{-2}$ s$^{-1}$) in comparison with LPS-EB alone, indicating that bindings involving IFI16 are much less stable once formed. In agreement with the cytokine release, the aforementioned results indicate that, when added separately, LPS-EB binds to TLR4/MD2 more rapidly than IFI16 alone. In this setting, LPS-EB *per se* is able to trigger a weak inflammatory response highly likely due to its slow dissociation from the receptor, which in turn delays the optimal turnover of the receptor. In contrast, IFI16/LPS-EB complex binds to TLR4 more rapidly than LPS-EB simultaneously added to the cells, and it is released much more rapidly.

## Discussion

We previously reported that extracellular IFI16 promotes IL-6 and IL-8 production in endothelial cells, and that such proinflammatory activity is amplified in the presence of subtoxic concentrations of LPS-EB, a full activator of the TLR4 signaling pathway [24]. Here, we expand on those observations by showing that, in renal and monocytic cell lines, IFI16 either alone or in complex with LPS binds to TLR4, thereby triggering a proinflammatory response through the TLR4/MD2/MyD88 signaling pathway. Specifically, by means of *in vitro* pull-down assays and saturation binding experiments, we provide the first evidence that the HINB domain of IFI16 mediates complex formation with LPS-EB or LPS-F583, two *E. coli*-derived variants of LPS capable of acting as strong TLR4 agonists [35,36]. This interaction follows a prototypical associative binding, with increasing rate of binding up to the plateau phase following addition of increasing amounts of the analytes. Furthermore, this binding is not dependent on the polysaccharide outer chain length as both LPS-EB and LPS-F583 display similar binding affinity for IFI16—the LPS-F583 variant is in fact characterized by the presence of a shorter polysaccharide chain compared to that of LPS-EB [37]. In addition, we show that both LPS-PG, a weak TLR4 agonist [38], and LPS-RS, a TLR4 antagonist [39]—these molecules display fewer acyl chains in their lipid A moieties compared to LPS-EB—, bind to IFI16 with similar affinities, albeit slightly lower than those of LPS-EB and LPS-F583. Finally, using the *E. coli* F583-derived DPLA and MPLA *lipid A variants* [37,40], we demonstrate that lipid A is the LPS moiety interacting with IFI16-HINB, affording the highest affinity for LPS. Accordingly, the detoxified variant of LPS-EB, containing a lipid A moiety partially delipidated by alkaline hydrolysis, binds weakly to IFI16. The observation that the HINB domain of IFI16 has a much higher affinity for lipid A than that of the HINA domain, despite both molecules being highly similar in terms of primary sequence and overall structure topology, is only partially unexpected. Indeed, these two IFI16 domains have already been shown to have distinct modes of binding to another paradigmatic PAMP—*i.e.*, viral DNA—most likely due to their different folding structures [41,42].

In recent years, mounting evidence has shown how TLRs, besides sensing exogenous microbial components, are also capable of recognizing endogenous material released during cellular injury, thereby promoting a non-microbial-induced inflammatory state known as sterile inflammation, which if not resolved can lead to severe acute and chronic inflammatory conditions [43–45]. Here, we propose that IFI16 might represent a novel trigger of sterile inflammation acting through the TLR4 signaling pathway. In particular, we show that exposure to recombinant IFI16 can induce IL-6, IL-8 and TNF-α transcriptional activation and release of these cytokines into the culture supernatants. This induction is strictly dependent on the presence of the TLR4/MD2 receptor complex and the MyD88 adaptor. By contrast, the

membrane-associated CD14 receptor seems to be only marginally involved in this signaling pathway given that undifferentiated THP-1 cells, displaying low levels of CD14 expression, and 786-O cells, expressing high levels of CD14, show similar cytokine induction patterns upon IFI16 exposure. The fact that IFI16 broadly activates inflammation through TLR4 signaling pathways strengthens the notion that extracellular IFI16 acts as a DAMP capable of promoting inflammation. Fittingly, aberrant IFI16 expression—*i.e.*, overexpression of IFI16 in otherwise negative cells or IFI16 delocalization to the cytoplasm—has been reported in a number of inflammatory conditions, such as SLE (skin) [8], psoriasis (skin) [11–13], SSc (skin) [10], IBD (colonic epithelium) [6,7] and SS (salivary epithelial and inflammatory infiltrating cells) [14,15,17]. Additionally, aberrant IFI16 expression has been reported in virus-infected cells [21–23] or cells treated with IFN-γ [46]. Importantly, in some of these and other pathological conditions, we and others have shown that IFI16 exists in a free, extracellular form in the blood or extracellular milieu [14–16,47]. Particularly, we found that high levels of circulating IFI16 (≥ 27 ng/ml) were associated to overall worse clinical parameters in three cohorts of RA, SS and PsA patients. Notably, among RA patients, circulating IFI16 was more frequently found in subjects with rheumatoid factor (RF)/anti-CCP-positive serum and significantly associated with pulmonary involvement [16]. Furthermore, in SS patients, circulating IFI16 is associated with increased prevalence of both RF and glandular infiltration degree [14], while in PsA patients is associated with elevated C-reactive protein (CRP) levels [19]. The release of extracellular IFI16 has also been shown by our group in a model of keratinocytes exposed to UVB radiation [8]. Although the biological rationale of these findings is far from being completely understood, these observations clearly indicate that the IFI16 protein, whose expression in the natural setting is restricted to the nuclei of a limited number of cell types, such as keratinocytes, fibroblasts, endothelial and hematopoietic cells [1], can be released by a broad spectrum of injured cells, including damaged epithelial cells or the inflammatory cells recruited at the site of injury, which are known to massively express IFI16. In this setting, as mentioned above, extracellular IFI16 can act as a DAMP in promoting sterile inflammation [48]. Accordingly, the exposure of THP-1 cells to conditioned medium from UVB-treated cells containing the IFI16 protein was able to significantly enhanced IL-6, IL-8 and TNF-α release when compared to the conditioned medium from UVB-treated IFI16 knockout cells. Addition of LPS-EB but not that of the weak TLR4-agonist LPS variant further enhanced cytokine induction, while pre-treatment of THP-1 cells with anti-TLR4 antibodies almost abolished the cytokine release. Thus, it is tempting to speculate that, similarly to pathogen-induced inflammation, binding of extracellularly-released IFI16 to TLR4 can activate both non-immune and innate immune cells, thus leading to the production of various cytokines and chemokines responsible for the recruitment of additional inflammatory cells [49].

In agreement with the emerging concept that DAMPs often potentiate their activity by binding to PAMPs, we demonstrate that the IFI16 proinflammatory activity is significantly enhanced when the protein is pre-incubated with subtoxic concentration of LPS and then added to the cells as pre-formed complex. Consistently, this effect is not observed when a truncated variant of IFI16 lacking the LPS-binding domain is used. Despite the fact that IFI16 binds with similar affinity to different variants of LPS, we could only achieve a significant increase in proinflammatory cytokine release with the strong TLR4 agonists LPS-EB and LPS-F583. Of note, these LPS molecules when added alone to the cells, even at high doses, were only able to induce marginally the transcriptional activation of such cytokines. Interestingly, the LPS-F583-derived lipid A DPLA and MPLA, carrying respectively a di- and a mono-phosphorylated glucosamine dimer, both lacking the sugar inner core, display a remarkably different ability to enhance IFI16 activity. Although both molecules show the highest affinity for IFI16 *in vitro*, only DPLA partially retains the ability to potentiate IFI16 downstream

signaling. Fittingly, IFI16 binding to *Rhodobacter sphaeroides*-derived LPS, which is known to antagonize the response to strong TLR4 activators in human and mouse monocytes [39], did not affect IFI16 proinflammatory activity. Interestingly, competition binding experiments in 786-O and THP-1 cells suggest that the affinity for TLR4 of either LPS-EB or LPS-RS is higher than that of IFI16. Indeed, when the cells are exposed to IFI16 and LPS-EB that were not pre-complexed, the release of proinflammatory cytokines in the culture supernatants is far lower than that of cells exposed to pre-complexed IFI16/LPS-EB or IFI16 alone. Accordingly, the binding kinetics revealed by SPR analysis clearly indicated that LPS-EB has a much higher affinity for TLR4 and a very slow kinetics of dissociation when compared to the IFI16 protein alone. Conversely, the IFI16/LPS-EB complex retains a higher affinity for TLR4 and is not displaced upon co-treatment with LPS-EB, as attested by the release of cytokines at levels similar to those observed in the supernatants of cells exposed to the complex alone. Likewise, IFI16/LPS-RS complex activity is not affected by the simultaneous addition of an equal amount of LPS-RS. In good agreement with the immunoprecipitation and competition assays, binding kinetics analysis by SPR reveals that LPS-EB, regardless of its overall higher affinity for TLR4/MD2, cannot compete with the IFI16/LPS-EB complex for binding to the receptor. Indeed, the IFI16/LPS-EB complex appears to be continuously engaged for TLR4 activation, as indicated by its faster association and dissociation rates. Thus, these data strongly suggest that *in vivo* i) the proinflammatory activity of IFI16 is enhanced upon its interaction with small amounts of circulating LPS, and ii) the IFI16/LPS-EB complex has a rapid stimulation turnover on the receptor, successfully competing with LPS-EB alone and leading to a massive inflammatory response (Fig 9).

Overall, our findings unveil a central role of extracellular IFI16 in triggering inflammation thanks to its ability to bind to the TLR4/MD2 complex, thereby triggering TLR4/MyD88/NF-κB signaling. Given that IFI16 is able to form stable complexes with various LPS variants through interaction of its HINB domain with the lipid A moiety of LPS, we propose a new pathogenic mechanism regulated by extremely fine-tuned interactions between extracellular IFI16 and subtoxic doses of LPS, which are known to be present in various pathological settings other than gram-negative infections [50–52].

## Materials and methods

### Reagents, antibodies, and recombinant proteins

LPS from *Escherichia coli* O111:B4 (LPS-EB), *Porphyromonas gingivalis* (LPS-PG) or *Rhodobacter sphaeroides* (LPS-RS), biotin-labeled LPS from *Escherichia coli* O111:B4 (biotin-labeled LPS-EB), detoxified LPS from *Escherichia coli* O111:B4 (detoxLPS) and polimixin B (PMB) were all purchased from InvivoGen. LPS from *Escherichia coli* F583 (LPS-F583), monophosphoryl lipid A from *Escherichia coli* F583 (MPLA), diphosphoryl lipid A from *Escherichia coli* F583 (DPLA) were purchased from Sigma-Aldrich. Bovine serum albumin Fraction V pH 7 (BSA) was purchased from Euroclone.

The following antibodies were used: rabbit polyclonal anti-IFI16 N-term and C-term (produced as described in [1], mAb anti-human TLR4 (sc-293072, Santa Cruz Biotechnologies), mAb anti-human TLR4 (sc-13593, Santa Cruz Biotechnologies), mAb anti-human TLR4 (mabg-htlr4, InvivoGen), rabbit polyclonal anti-MD2 (AHP1717T, Bio-Rad), rabbit monoclonal anti-MyD88 (4283, Cell Signaling Technology), mAb anti-NF-κB p65 (sc-8008 X, Santa Cruz Biotechnologies), PE mouse anti-human CD14 (555398, BD Pharmingen), mAb anti-β-actin (A1978, Sigma-Aldrich), rabbit IgG-HRP (A6154, Sigma-Aldrich,), mouse IgG-HRP (NA931V, GE Healthcare), streptavidin-HRP (E2886, Sigma-Aldrich,), mouse IgG-Alexa Flour 488 (A11001, Thermo Fisher Scientific), normal mouse IgG2a isotype control (sc-3878, Santa Cruz Biotechnologies).

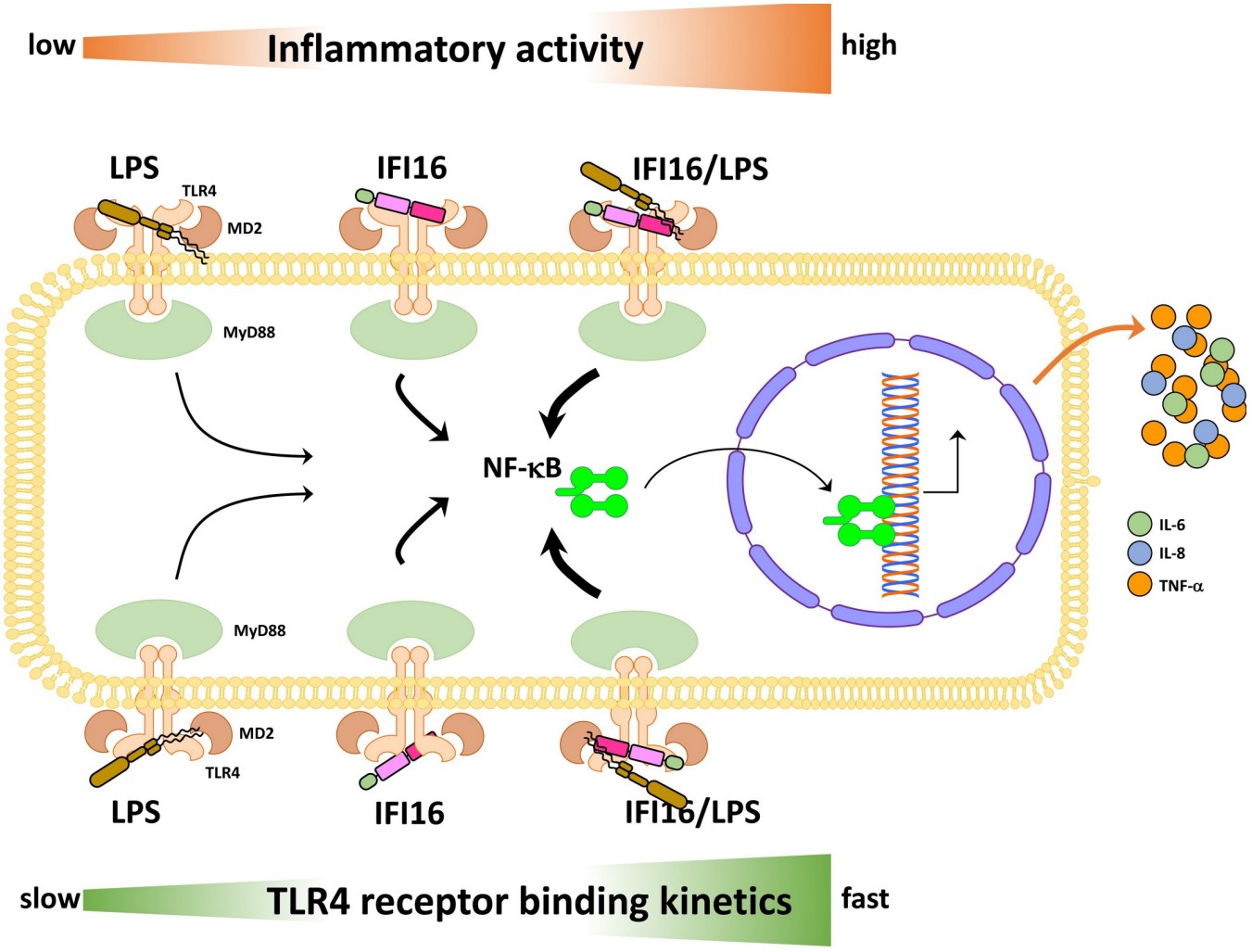

**Fig 9. Proposed model depicting the inflammatory activities and binding kinetics to TLR4 of LPS and IFI16, alone or in combination.** The relative inflammatory activities, from low to high, are reported in the upper part of the scheme (orange arrow). The relative binding kinetics to TLR4 are reported in the lower part of the scheme (green arrow). The thickness of the black arrows is directly proportional to the ability of the pathway to induce NF-κB activation.

Human recombinant IFI16 was produced as previously described [24]. The purified protein was then processed with Toxin Eraser Endotoxin Removal Kit (GenScript) to remove endotoxins, and the final endotoxin concentration was measured using Toxin Sensor Chromogenic LAL Endotoxin Assay Kit (GenScript). The endotoxin level was always below 0.05 EU/ml. Purified IFI16 was stored at—80˚C in endotoxin-free vials.

The coding regions of the three IFI16 domains (*i.e.*, PYRIN, HINA and HINB) and of the IFI16 variant lacking the HINB domain (IFI16ΔHINB) were amplified from full-length human IFI16 cDNA (isoform b) and cloned in pET30a expression vector (Novagen). The three domains and IFI16ΔHINB were then synthesized and processed following the same procedure as that for the full-length protein [24].

GST recombinant protein was expressed using pGEX-4T2 vector and purified according to standard procedures. The purity of the proteins was assessed by 12% SDS-polyacrylamide gel electrophoresis. Recombinant TLR4 protein and TLR4/MD2 complex (478-TR-050 and 3146-TM-050/CF, respectively) were purchased from R&D Systems.

## Pull-down assay, ELISA and competitive ELISA

Biotin-labeled LPS-EB (10 µg) was incubated with 30 µl of streptavidin sepharose high performance beads (GE Healthcare) for 3 h at 4˚C. After a washing step, 3 µg of recombinant IFI16, PYRIN, HINA, HINB, IFI16ΔHINB, or GST were added and incubated O/N at 4˚C. After five washes with 1X PBS with 0.25% Triton X-100 (Sigma-Aldrich), samples were boiled in sample buffer containing SDS and β-mercaptoethanol and centrifuged. Supernatants were separated on a 7.5% or 12% SDS-polyacrylamide gel (Bio-Rad). Gels were stained with Coomassie brilliant blue (Serva Electrophoresis GmbH) for protein visualization.

For saturation binding experiments, microtiter plates (Nunc-Immuno MaxiSorp, Thermo Fischer Scientific) were coated with 2 µg/ml of recombinant IFI16 or 10 µg/ml of BSA or GST in 1X PBS O/N at 4˚C. After a washing step with 1X PBS and 0.25% Tween 20 (v/v, Sigma-Aldrich) and blocking step with 1X PBS with 3% BSA and 0.05% Tween 20 for 1 h, increasing concentrations of biotin-labeled LPS-EB, preincubated with 10 µg/ml of polymyxin B (PMB) when specified, were added to the wells and incubated for 1 h at room temperature (RT). Bound proteins were then detected using HRP-conjugated streptavidin. TMB solution (Thermo Fischer Scientific) was used for color development, and OD was measured at 450 nm. Alternatively, microtiter plates were coated with 10 µg/ml of LPS-EB, LPS-PG, LPS-RS, LPS-F583, MPLA, DPLA, or detoxLPS in 1X PBS for 24 h at RT. After washing and blocking for 2 h, increasing concentrations of IFI16, PYRIN, HINA, or HINB were added to the wells, preincubated with 10 µg/ml of PMB when specified, for 2 h. Anti-IFI16 antibodies against the N- or the C-terminus of the protein and HRP-labeled anti-rabbit IgG were then added as primary and secondary antibodies, respectively. The binding was detected as described above.

For whole-cell ELISA, different strains of bacteria (*i.e.*, gram-positive: *Staphylococcus aureus*, *Staphylococcus epidermidis* and *Streptococcus pyogenes*; gram-negative: *Escherichia coli* and *Klebsiella pneumonia*) were grown in LB medium without antibiotics and, after washing with 1X PBS, fixed in 0.5% formalin O/N at 4˚C. Subsequently, the bacteria were diluted to an $OD_{600}$ of 0.5 and were used to coat microtiter plates O/N at 37˚C. After blocking, increasing concentrations of IFI16 were added to the wells and incubated for 2 h at RT. Anti-IFI16 antibodies against the N-terminus of the protein and HRP-labelled anti-rabbit IgG were then added as primary and secondary antibody, respectively, and binding was detected as described above.

For competitive ELISA, microtiter plates were coated with 1 µg/ml LPS-EB in 1X PBS O/N at RT. Successively, a constant amount of 2 µg/ml IFI16 was added to the wells in the presence of increasing concentration of LPS-EB, MPLA or detoxLPS. After incubation for 4 h at RT under gentle agitation, plates were incubated with an anti-IFI16 N-terminal primary antibody and an HRP-conjugated anti-rabbit IgG secondary antibody. The binding was detected as described above. To determine $K_D$ constants, saturation binding experiments were performed, and data were fitted to the Langmuir isotherm equation, which describes the equilibrium binding of the ligands [53]. Data are reported as sigmoid concentration-response curves plotted against log concentrations.

## Cell cultures, treatments and transfection

Human kidney adenocarcinoma cells 786-O and human leukemia monocytes THP-1 were obtained from ATCC and grown in RPMI 1640 Medium (Sigma-Aldrich) containing 10% of fetal bovine serum (FBS, Immunological Sciences) and 1% of penicillin/streptomycin/glutamine solution (PSG, Gibco) at 37˚C and 5% $CO_2$. Wild-type and IFI16-knockout (U2OS-IFI16$^{-/-}$) human osteosarcoma cells U2OS were kindly gifted by Dr. Bala Chandrani-versity of South Florida, FL, USA) [22] and grown in Dulbecco's modified Eagle's medium

(Sigma-Aldrich) containing 10% of FBS and 1% of PSG at 37°C and 5% $CO_2$. UVB irradiations were performed as previously described [8]. The resulting cell culture supernatants were centrifuged to remove any cellular pellet and stored at -80°C for the following experiments.

For treatments, cells were stimulated in complete medium with endotoxin-free IFI16 (25 μg/ml), endotoxin-free IFI16ΔHINB (25 μg/ml), MPLA, DPLA, LPS-F583, LPS-EB, LPS-RS, alone or pre-complexed by O/N incubation at 4°C, unless specified otherwise. Additionally, cells were stimulated with supernatants of untreated or UVB-treated U2OS or U2OS-IFI16$^{-/-}$ cells alone or preincubated O/N at 4°C with LPS-EB, or LPS-RS. LPS variants or lipid A moieties were used at a concentration of 10 ng/ml. All treatments were carried out at 37°C and 5% $CO_2$.

For TLR4 neutralization, THP-1 cells were pretreated with 10 μg/ml of anti-TLR4 antibodies for 1 h before treatments.

For TLR4, MD2 or Myd88 gene silencing, cells were transfected with specific human TLR4, MD2, Myd88 or control siRNAs (Dharmacon, siGENOME smart pool) using DharmaFect1 transfection reagent (Dharmacon). The efficiency of knockdown was confirmed by FACS analysis and immunoblotting at 48 h after transfection.

## FACS analysis

Single cell suspensions were incubated for 30 min on ice with anti-TLR4 (sc-13593), PE-conjugated anti-CD14 (555398) or with isotype control diluted in staining buffer (PBS 1% FBS 0.1% $NaN_3$). To detect TLR4 staining, cells were further washed and incubated for 30 min on ice with Alexa Fluor 488-conjugated secondary antibody. Cell counts and fluorescence intensity measurements were calculated by Attune NxT Flow Cytometer (Thermo Fisher Scientific). Background fluorescence was subtracted using unlabeled cells, and channel compensation was performed using Attune performance tracking beads (Thermo Fisher Scientific). A total of 10,000 events were recorded. Data were analyzed by FlowJo cell analysis software (BD Life Sciences).

## Western blot and immunoprecipitation

Whole-cell extracts were prepared using RIPA lysis and extraction buffer (Pierce) with halt protease and phosphatase inhibitor (Thermo Fisher Scientific) on ice, and total protein concentration was quantified by Bradford Reagent (Sigma-Aldrich) measuring absorbance at 595 nm. Twenty μg of cell extracts, or 30 μl of U2OS culture supernatants were separated by electrophoresis on 7.5% or 12% SDS-polyacrylamide gels (Bio-Rad), transferred to nitrocellulose membranes, blocked with 10% non-fat milk in tris-buffered saline-tween (TBST), and probed with specific primary antibodies O/N at 4°C. After being washed with TBST, membranes were incubated with specific HRP-conjugated secondary antibodies, and binding was detected by ECL (Thermo Fisher Scientific, Super Signal West Pico). Expression of β-actin was used as protein loading control.

Co-immunoprecipitation of TLR4 with interacting proteins was performed using the Dynabeads Protein G Immunoprecipitation Kit (ThermoFisher), according to the manufacturer's instructions with minor modifications. Briefly, after lysis of treated cells, 20 μg of total cell extracts were kept as the input control, while 90 μg of total cell extracts were incubated for 1 h at RT with 2.5 μg of anti-TLR4 antibody previously conjugated with magnetic beads. The resulting complexes were then washed, eluted, denatured, and subjected to Western blotting as described above. For DNase-treated cell extracts, DNase I (Sigma Aldrich) was added at a 1:10 dilution and incubated for 15 min at RT. Images were acquired, and densitometry of the bands was performed using Quantity One software (version 4.6.9, Bio-Rad). Densitometry values were normalized using the corresponding loading controls.

## Quantitative real time PCR

Quantitative real-time PCR (qRT-PCR) was performed on a CFX96 Real-Time PCR Detection System (Bio-Rad) as previously described [54]. Total RNA was extracted using TRI Reagent (Sigma-Aldrich), and 1 μg was retrotranscribed using an iScript cDNA Synthesis Kit (Bio-Rad). Reverse-transcribed cDNAs were amplified in duplicate using SsoAdvanced Universal SYBR Green Supermix (Bio-Rad). The glyceraldehyde 3-phosphate dehydrogenase (GAPDH) gene was used as housekeeping gene to normalize for variations in cDNA levels. The relative normalized expression after stimulation as compared to control was calculated as fold change = $2^{-\Delta(\Delta CT)}$ where $\Delta CT = CT_{target} - CT_{GADPH}$ and $\Delta(\Delta CT) = \Delta CT_{stimulated} - \Delta CT_{control}$. The primer sequences are available upon request.

## Cytokines measurement by ELISA

Cytokines secreted in culture supernatants after treatments were analyzed using human IL-6 DuoSet ELISA and human IL-8 DuoSet ELISA (all from R&D Systems), human TNF-α Uncoated ELISA and human IL-1β Uncoated ELISA (Thermo Fisher Scientific) according to the manufacturer's instructions. Absorbance was measured using a Spark multimode micro-plate reader (Tecan).

## Transcription factor assay

Nuclear extracts were prepared using NE-PER Nuclear and Cytoplasmic Extraction Reagents (Thermo Fisher Scientific), according to the manufacturer's instructions. NF-κB binding activity to a DNA probe containing its binding consensus sequence was measured by Universal Transcription Factor Assay Colorimetric kit (Merck Millipore). The binding of NF-κB to the DNA probe was revealed using a specific primary antibody, with an HRP-conjugated secondary antibody used for detection with TMB substrate. The intensity of the reaction was measured at 450 nm. The following biotinylated oligonucleotides were used: sense (biotin): 5'-AT GACATAGGAAAACTGAAAGGGAGAAGTGAAAGTGGGAAATTCCTCTG-3'; antisense: 5'-CAGAGGAATTTCCCACTTTCACTTCTCCCTTTCAGTTTTCCTATGTCAT-3'.

## Surface plasmon resonance analysis

The Biacore X100 (GE Healthcare) instrument was used for real-time binding interaction experiments. Recombinant TLR4 or TLR4/MD2 complex was covalently immobilized onto the surface of sensor CM5 (cat # BR100012, GE Healthcare) chips *via* amine coupling. TLR4 was diluted to a concentration of 10 μg/ml in 10 mM sodium acetate at pH 4.0, while TLR4/MD2 complex was diluted to a concentration of 20 μg/ml in the same buffer. Both proteins were injected on CM5 chips at a flow rate of 10 μl/min, upon activation of the carboxyl groups on the sensor surface with 7-min injection of a mixture of 0.2 M EDC and 0.05 M NHS. The remaining esters were blocked with 7-min injection of ethanolamine. Taking into account the ligands (TLR4 or TLR4/MD2) and analytes (IFI16, LPS-EB or IFI16/LPS-EB) molecular weights (MW) of 70 or 90 kDa, and 90, 10 or 100 kDa respectively, the appropriate ligand density (RL) on the chip was calculated according to the following equation: RL = (ligand MW/ analyte MW) × Rmax × (1/Sm), where Rmax is the maximum binding signal and Sm corresponds to the binding stoichiometry. The target capture level of the TLR4 or TLR4/MD2 was of 596.0 or 1,223.9 response units (RUs), respectively. The other flow cell was used as a reference and was immediately blocked after the activation. Increasing concentrations of endotoxin-free IFI16, LPS-EB or IFI16/LPS-EB complex were flowed over the CM5 sensor chip coated with TLR4 or TLR4/MD2 at a flow rate of 30 μl/min at 25°C with an association time of

120 s for IFI16 alone and the IFI16/LPS-EB complex, and 180 s for LPS-EB, and a dissociation phase of 180 s for IFI16 and IFI16/LPS-EB complex or 600 s for LPS-EB. A single regeneration step with 50 mM NaOH was performed following each analytic cycle. All the analytes tested were diluted in the HBS-EP+ buffer (GE Healthcare).

Recombinant IFI16 was covalently immobilized onto the surface of sensor CM5 chips via amine coupling as done for TLR4 and TLR4/MD2 complex. IFI16 was diluted to a concentration of 25 μg/ml in 10 mM sodium acetate at pH 4.0. The target capture level of IFI16 was of 1,926.6 response units (RUs). Increasing concentrations of LPS-EB, diluted in HBS-EP+ buffer, were flowed over the CM5 sensor chip coated with IFI16 at a flow rate of 30 μl/min at 25°C with an association time of 180 s and a dissociation phase of 600 s. A single regeneration step with 50 mM NaOH was performed following each analytic cycle. The $K_D$s were evaluated using the BIAcore evaluation software (GE Healthcare) and the reliability of the kinetic constants calculated by assuming a 1:1 binding model supported by the quality assessment indicators values.

## Statistical analysis

All statistical analyses were performed using GraphPad Prism version 6.00 for Windows (GraphPad Software, La Jolla California USA, www.graphpad.com). The data are expressed as mean ± SD. For comparisons between two groups, means were compared using a two-tailed Student's t test. For comparisons among three groups, means were compared using one-way or two-way ANOVA followed by Dunnett's test. Differences were considered statistically significant at a $P$ value < 0.05.

## Supporting information

**S1 Fig. IFI16 variant lacking HINB domain does not bind to LPS. (A)** Domain organization of the IFI16ΔHINB protein. The numbers represent the amino acid positions based on NCBI Reference Sequence NP_005522. From the N- to the C-terminal (left to right), IFI16ΔHINB comprises a pyrin domain, and only one hematopoietic interferon-inducible nuclear protein with 200-amino-acid repeats (HINA) domain. S/T/P = serine/threonine/proline-rich repeats. **(B)** Coomassie brilliant blue staining of pull-down assays performed with 3 μg of recombinant IFI16ΔHINB, full-length IFI16, or HINB domain in the presence or absence of biotin-labeled LPS from *E. coli* O111:B4 (biotin-LPS-EB).
(TIF)

**S2 Fig. Expression of TLR4 signaling molecules in 786-O and THP-1 cells. (A)** Western blot analysis of TLR4, MD2 and MyD88 in whole-cell lysates of 786-O cells or THP-1 cells. Immunoblot with anti-β-actin antibody was used as loading control. **(B)** Cell surface expression of TLR4 and CD14 in 786-O and THP-1 cells (left and right panels, respectively), detected by flow cytometry using specific antibodies. Blue histograms represent background fluorescence; red histograms denote TLR4 (upper panels) or CD14 (lower panels) staining. The y-axis represents the number of cells, while the x-axis represents the level of fluorescence (FL-2) in a logarithmic scale. Images are representative of two independent experiments with similar results. The percentage of stained cells is reported in each panel. MFI = mean fluorescence intensity.
(TIF)

**S3 Fig. Assessment of TLR4, MD2 and MyD88 knockdown upon siRNA transfection. (A)** Upper panel: Western blot analysis of TLR4 and β-actin in whole-cell lysates of 786-O cells or THP-1 cells transfected with non-targeting siRNA control (siCTRL) or siRNA TLR4 (siTLR4) at 48 h after transfection. Lower panel: densitometric analysis showing the fold change

expression of the indicated proteins expressed as the mean from three independent experiments. Error bars indicate SD. **(B)** Cell surface expression of TLR4 in 786-O and THP-1 cells (upper and lower panels, respectively), detected by flow cytometry using specific antibodies at 48 h after transfection with siCTRL or siTLR4. Blue histograms represent background fluorescence; red histograms denote TLR4 staining. The y-axis represents the number of cells, while the x-axis represents the level of fluorescence (FL-2) in a logarithmic scale. Images are representative of two independent experiments with similar results. The percentage of stained cells is reported in each panel. MFI = mean fluorescence intensity. **(C, D)** Upper panels: Western blot analysis of MD2 **(C)** and MyD88 **(D)** in whole-cell lysates from 786-O cells or THP-1 cells transfected with non-targeting siRNA control (siCTRL) or specific siRNAs—siMD2 and siMyD88, respectively—at 48 h after transfection. Lower panels: densitometric analysis showing the fold change expression of the indicated proteins expressed as the mean from three independent experiments. Error bars indicate SD.
(TIF)

**S4 Fig. Quantitative RT-PCR analysis of proinflammatory cytokine expression in TLR4, MD2 and MyD88 knockdown cells. (A-C)** qRT-PCR analysis of IL-6, IL-8 and TNF-α mRNA expression levels in 786-O or THP-1 cells transfected for 48 h with scramble control (siCTRL), or siRNAs against TLR4 (siTLR4) **(A)**, MD2 (siMD2) **(B)** or MyD88 (siMyD88) **(C)**. Cells were then stimulated for 24 h with IFI16 (25 μg/ml), LPS from *E. coli* O111:B4 (LPS-EB, 10 ng/ml) or IFI16/LPS-EB complex (preincubated O/N at 4°C), or left untreated (mock). Values were normalized to GAPDH mRNA and plotted as fold induction over mock-treated cells. qRT-PCR data are expressed as mean values of biological triplicates. Error bars indicate SD ($^{*}P < 0.05$, $^{**}P < 0.01$, $^{***}P < 0.001$, ns: not significant; unpaired Student's *t*-test for comparison of silenced cells *vs*. their relative control counterpart).
(TIF)

## Acknowledgments

We thank Marcello Arsura for critically reviewing the manuscript.

## Author Contributions

**Conceptualization:** Andrea Iannucci, Santo Landolfo, Marisa Gariglio, Marco De Andrea.

**Data curation:** Andrea Iannucci, Valeria Caneparo, Stefano Raviola, Gloria Griffante.

**Formal analysis:** Andrea Iannucci, Valeria Caneparo, Donato Colangelo, Riccardo Miggiano, Marco De Andrea.

**Funding acquisition:** Santo Landolfo, Marisa Gariglio, Marco De Andrea.

**Investigation:** Andrea Iannucci, Valeria Caneparo, Stefano Raviola, Isacco Debernardi, Gloria Griffante.

**Methodology:** Andrea Iannucci, Marisa Gariglio, Marco De Andrea.

**Project administration:** Santo Landolfo, Marisa Gariglio, Marco De Andrea.

**Resources:** Santo Landolfo, Marisa Gariglio, Marco De Andrea.

**Supervision:** Marisa Gariglio, Marco De Andrea.

**Validation:** Andrea Iannucci, Donato Colangelo, Riccardo Miggiano, Marisa Gariglio, Marco De Andrea.

**Visualization:** Andrea Iannucci, Marco De Andrea.

**Writing – original draft:** Andrea Iannucci, Marisa Gariglio, Marco De Andrea.

**Writing – review & editing:** Andrea Iannucci, Donato Colangelo, Riccardo Miggiano, Santo Landolfo, Marisa Gariglio, Marco De Andrea.

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
