## [Decision Letter · Decision Letter 0]

27 Mar 2020

Dear Prof. De Andrea,

Thank you very much for submitting your manuscript "Toll-like receptor 4-mediated inflammation triggered by extracellular IFI16 is enhanced by lipopolysaccharide binding" for consideration at PLOS Pathogens. As with all papers reviewed by the journal, your manuscript was reviewed by members of the editorial board and by several independent reviewers. In light of the reviews (below this email), we would like to invite the resubmission of a significantly-revised version that takes into account the reviewers' comments. In particular the reviewers expressed concerns about the artificiality of the experimental system utilized and the overall magnitude of responses observed.  

We cannot make any decision about publication until we have seen the revised manuscript and your response to the reviewers' comments. Your revised manuscript is also likely to be sent to reviewers for further evaluation.

Sincerely,

Victor Robert DeFilippis

Guest Editor

PLOS Pathogens

Klaus Früh

Section Editor

PLOS Pathogens

Kasturi Haldar

Editor-in-Chief

PLOS Pathogens

orcid.org/0000-0001-5065-158X

Michael Malim

Editor-in-Chief

PLOS Pathogens

orcid.org/0000-0002-7699-2064

Reviewer's Responses to Questions

**Part I - Summary**

Reviewer #1: In the present study, the authors followed up on their previous observation that IFI16 is detectable in sera from immune-activated individuals and that recombinant IFI16 boosts inflammatory responses in cell culture. Here, they examined the underlying mechanism of this effect of IFI16. They present evidence that recombinant IFI16 binds to LPS on the surface of Gram- bacteria, regardless of their potency in activating TLR4 signaling. In addition, they found that the lipid A component of LPS and the HIN-B domain of IFI16 play important roles in this interaction. Finally, the authors further show that IFI16 complexed with the LPS displays a slightly higher ability than IFI16 alone to trigger immune activation via the TRAP/MyD88/NF-κB pathway. Finally, Iannucci and colleagues report that IFI16 directly binds to TLR4 and that the binding affinity of the IFI16/LPS complex is higher compared to the LPS alone. In conclusion, they propose that IFI16 may be a novel DAMP that potentiates the action of TLR4 by promoting its signaling by binding to subtoxic doses of LPS.

The study is competently performed and the results potentially interesting. The major limitation of the present (and previous) study is that all results were obtained using bacterial expressed tagged IFI16. Usually, IFI16 shows a nuclear localization and it is difficult to assess how the concentrations of circulating IFI16 achieved in vivo compared to those used in the present study. In addition, the possibility that endogenous and bacterial expressed recombinant IFI16 show structural and functional differences cannot be excluded.

Reviewer #2: This study shows that IFI16 is a DAMP that activates inflammatory response through TLR4. In addition if binds to small amounts of LPS through the HIN-B domain enhancing LPS capacity to activate TLR4

The study is original, novel and extremely well organized. Biochemical experiments are extensive and convincing. The paper is well-written and well referenced

Reviewer #3: In this study, the authors suggest that IFI16 interacts with LPS and TLR4 to promote inflammatory cytokine expression. This idea is interesting and the authors provide some interesting biochemical data to support the interactions proposed. However, the functional significance of these interactions (presented in Figure 3) are of such minor significance that I am not convinced by the conclusions offered. The data presented shows very modest effects of IFI16 on LPS signaling (less than two fold). Whether a 2 fold increase in LPS signaling is of importance in any physiological setting is undefined. For these reasons, I cannot support the conclusions of this study. Much more mechanistic and functional analysis in vitro and in vivo would need to be performed before the model proposed could be considered solid enough for publication. One minor comment is indicated below.

1. Line 75 states that LPB transfers LPS to CD14. This statement is incorrect. Elegant studies from Weiss and colleagues demonstrated that LBP binds to LPS and enables CD14 to extract a monomer of this PAMP. LBP does not transfer LPS, it allows CD14 to extract LPS. Please correct this statement and cite the appropriate publications.

**Part II – Major Issues: Key Experiments Required for Acceptance**

Reviewer #1: The results are potentially interesting. However, some evidence that these interactions and effects can be observed using IFI16 released from immune cells seems required to ensure that the results are relevant and not just representing an in vitro artifact of overexpressed recombinant agents. Some more minor issues are specified below.

Analysis of individual domains of IFI16 supports that the HIN-B domain is important for the recognition of LPS. To verify this, they should show that IFI16 lacking the HIN-B domain does not interact with LPS to promote cytokine production.

Can the authors exclude that DNA has an impact on the outcome of the Co-IPs? It has been reported that both LPS and IFI16 may bind to DNA.

Reviewer #2: The Biochemical experiment are flawless and very convincing

Reviewer #3: see above

**Part III – Minor Issues: Editorial and Data Presentation Modifications**

Reviewer #1: Fig. 2E: lanes should be numbered.

The quantities of IFI16 and LPS used for complex formation are very different (i.e. 25 µg/ml and 25 ng/ml). Stoichiometries should be discussed. In this regards it would also be more informative to provide µM rather than µg/ml.

More detail should be provided on the conditions and kinetics of complex formation.

It seems that Figs. 1E and 5D are not mentioned in the text.

Fig. 6 A-B, IL-6 panel: stimulation of THP-1 cells with IFI16 with either DPLA or MPLA shows only a modest effect of IL-6 gene expression but clearly increases release of this cytokine. This should be discussed.

Fig. 7 A-B: It should be specified to which significant differences the stars are referring. When IFI16 and LPS-EB were added to the cells without pre-incubation, cytokine induction was lower than that measured with IFI16. Does this mean free LPS inhibits the effect of IFI16?

Reviewer #2: The authors show that LPS binds the HIN-B domain of IFI16 but do not elaborate on why only this domain can bind. What is the difference between the NIH-A and the HIN-B domains? Is there any amino acid motif that could be responsible for binding LPS? Can this motif be found in any other protein of the same family (ALRs?)

Is there any polymorphism in this motif that is associated to inflammatory or infectious disease in the population? The authors should expand their speculations in the discussion and propose scenarios in which disruption of this interaction may relate to human diseases

Reviewer #3: (No Response)

PLOS authors have the option to publish the peer review history of their article (what does this mean?). If published, this will include your full peer review and any attached files.

Reviewer #1: No

Reviewer #2: No

Reviewer #3: No
---

## [Editor Report · Decision Letter 1]

7 Jul 2020

Dear Prof. De Andrea,

Thank you very much for submitting your manuscript "Toll-like receptor 4-mediated inflammation triggered by extracellular IFI16 is enhanced by lipopolysaccharide binding" for consideration at PLOS Pathogens. As with all papers reviewed by the journal, your manuscript was reviewed by members of the editorial board and by several independent reviewers. The reviewers appreciated the attention to an important topic. Based on the reviews, we are likely to accept this manuscript for publication, providing that you modify the manuscript according to the review recommendations.

1. Page 11 states that IFNb was not detected when cells were stimulated with LPS but this is contrary to what's known about LPS/TLR4 signaling. Please provide an explanation for this anomaly.

2. Figure 5A x-axis labels read "siTRL4". Please correct.

Sincerely,

Victor Robert DeFilippis

Guest Editor

PLOS Pathogens

Klaus Früh

Section Editor

PLOS Pathogens

Kasturi Haldar

Editor-in-Chief

PLOS Pathogens

orcid.org/0000-0001-5065-158X

Michael Malim

Editor-in-Chief

PLOS Pathogens

orcid.org/0000-0002-7699-2064

1. Page 11 states that IFNb was not detected when cells were stimulated with LPS but this is contrary to what's known about LPS/TLR4 signaling. Please provide an explanation for this anomaly.

2. Figure 5A x-axis labels read "siTRL4". Please correct.
---

## [Editor Report · Decision Letter 2]

14 Jul 2020

Dear Prof. De Andrea,

We are pleased to inform you that your manuscript 'Toll-like receptor 4-mediated inflammation triggered by extracellular IFI16 is enhanced by lipopolysaccharide binding' has been provisionally accepted for publication in PLOS Pathogens.

Best regards,

Victor Robert DeFilippis

Guest Editor

PLOS Pathogens

Klaus Früh

Section Editor

PLOS Pathogens

Kasturi Haldar

Editor-in-Chief

PLOS Pathogens

orcid.org/0000-0001-5065-158X

Michael Malim

Editor-in-Chief

PLOS Pathogens

orcid.org/0000-0002-7699-2064
---

## [Editor Report · Acceptance letter]

4 Sep 2020

Dear Prof. De Andrea,

We are delighted to inform you that your manuscript, "Toll-like receptor 4-mediated inflammation triggered by extracellular IFI16 is enhanced by lipopolysaccharide binding," has been formally accepted for publication in PLOS Pathogens.

Best regards,

Kasturi Haldar

Editor-in-Chief

PLOS Pathogens

orcid.org/0000-0001-5065-158X

Michael Malim

Editor-in-Chief

PLOS Pathogens

orcid.org/0000-0002-7699-2064